# DDK regulates replication initiation by controlling the multiplicity of Cdc45-GINS binding to Mcm2-7

Lorraine De Jesús-Kim[1], Larry J Friedman[2], Marko Lõoke[1†], Christian K Ramsoomair[1], Jeff Gelles[2]*, Stephen P Bell[1]*

[1]Howard Hughes Medical Institute, Department of Biology, Massachusetts Institute of Technology, Cambridge, United States; [2]Department of Biochemistry, Brandeis University, Waltham, United States

**Abstract** The committed step of eukaryotic DNA replication occurs when the pairs of Mcm2-7 replicative helicases that license each replication origin are activated. Helicase activation requires the recruitment of Cdc45 and GINS to Mcm2-7, forming Cdc45-Mcm2-7-GINS complexes (CMGs). Using single-molecule biochemical assays to monitor CMG formation, we found that Cdc45 and GINS are recruited to loaded Mcm2-7 in two stages. Initially, Cdc45, GINS, and likely additional proteins are recruited to unstructured Mcm2-7 N-terminal tails in a Dbf4-dependent kinase (DDK)-dependent manner, forming Cdc45-tail-GINS intermediates (CtGs). DDK phosphorylation of multiple phosphorylation sites on the Mcm2-7 tails modulates the number of CtGs formed per Mcm2-7. In a second, inefficient event, a subset of CtGs transfer their Cdc45 and GINS components to form CMGs. Importantly, higher CtG multiplicity increases the frequency of CMG formation. Our findings reveal the molecular mechanisms sensitizing helicase activation to DDK levels with implications for control of replication origin efficiency and timing.

**\*For correspondence:**
gelles@brandeis.edu (JG);
spbell@mit.edu (SPB)

**Present address:** †Institute of Technology, University of Tartu, Tartu, Estonia

**Competing interests:** The authors declare that no competing interests exist.

## Introduction

Eukaryotic DNA replication coordinates the assembly of multi-protein replisomes at origins of replication to ensure complete genome duplication. Replisome assembly begins during G1 phase when two copies of the Mcm2-7 helicase are loaded around origin DNA as a head-to-head double hexamer (*Yardimci and Walter, 2014*). In S phase, the association of helicase-activating factors selects a subset of these helicase double hexamers to initiate DNA unwinding and form the core of a bidirectional pair of replisomes (*Mantiero et al., 2011*; *Tanaka et al., 2011*). Correct temporal control of helicase activation both ensures that the genome is replicated exactly once per cell cycle (*Remus and Diffley, 2009*) and reserves a subset of origins to complete replication even if replisomes derived from adjacent origins stall (*Blow et al., 2011*; *Mantiero et al., 2011*).

Helicase activation is the committed step of replication initiation and is controlled during the cell cycle by two kinases: S-CDK and Dbf4-dependent kinase (DDK) (*Figure 1—figure supplement 1*; *Labib, 2010*). DDK phosphorylates the long unstructured N-terminal tails of Mcm4 and Mcm6 driving the recruitment of Sld3/7 and Cdc45 to Mcm2-7 double hexamers (*Kamimura et al., 2001*; *Kanemaki and Labib, 2006*; *Sheu and Stillman, 2010*; *Heller et al., 2011*; *Randell et al., 2010*; *Deegan et al., 2016*). S-CDK phosphorylates Sld2 and Sld3/7, leading them to bind to two distinct pairs of BRCT repeats in Dpb11 (*Tanaka et al., 2007*; *Zegerman and Diffley, 2007*; *Muramatsu et al., 2010*). These S-CDK-dependent interactions recruit GINS to Sld3/7 associated with loaded Mcm2-7 complexes. Although both Cdc45 and GINS recruitment involve Sld3/7, whether these proteins are recruited by the same or different Sld3/7 molecules is unknown. Cdc45 and GINS are direct activators of the Mcm2-7 complex, and association of these factors with the

core structured region of Mcm2-7 is required to form a pair of Cdc45-Mcm2-7-GINS complexes (CMGs), the active eukaryotic replicative helicase (*Ilves et al., 2010*; *Moyer et al., 2006*; *Pacek et al., 2006*; *Yuan et al., 2016*; *Goswami et al., 2018*). Finally, Mcm10 activates CMG complexes to unwind origin DNA (*van Deursen et al., 2012*; *Kanke et al., 2012*; *Lõoke et al., 2017*; *Douglas et al., 2018*). The result of this process is the formation of two oppositely directed, active CMG complexes that generate the ssDNA necessary to recruit the rest of the replisome (*Bell and Labib, 2016*).

Although clear in broad strokes, multiple elements of CMG formation and helicase activation remain poorly understood. For example, current studies suggest that there are many Sld3-binding sites on the N-terminal tails of Mcm4 and Mcm6, although a specific binding motif has not been defined (*Deegan et al., 2016*). However, whether one or multiple sites are used during a given helicase-activation event is unknown. It is also unclear whether Cdc45 recruitment to the two Mcm2-7 complexes in the double hexamer is concerted. The possibility of coordinated Cdc45 recruitment is raised by evidence suggesting that Sld3 and Sld7 form a heterotetramer that could facilitate simultaneous recruitment of two Cdc45s (*Itou et al., 2015*). Lastly, the levels of DDK and a subset of helicase-activation proteins are known to modulate the timing and efficiency of helicase activation (*Mantiero et al., 2011*; *Tanaka et al., 2011*). However, the attributes of helicase activation that sensitizes this process to the levels of these proteins are not understood. To answer these questions and investigate the kinetics of the events leading to helicase activation, we developed a single-molecule (SM) method to directly observe this process in real time at a defined origin of replication in vitro.

SM biochemical techniques are well equipped to address questions of kinetics and stoichiometry during multi-protein complex assembly events like formation of the eukaryotic replisome (*Stratmann and van Oijen, 2014*). Previous SM studies have complemented both structural and biochemical studies on helicase loading (*Ticau et al., 2015*; *Ticau et al., 2017*; *Champasa et al., 2019*). Like most complex multi-protein assembly processes, ensemble helicase-activation reactions occur asynchronously due to the large number of protein–DNA and protein–protein interactions required. In addition, these bulk assays are typically performed as end-point assays with limited kinetic information (*Yeeles et al., 2015*), preventing detection of transient events. In contrast, SM biochemical studies allow post-hoc synchronization of asynchronous events permitting detailed kinetic analysis. In addition, by allowing real-time visualization, SM approaches can detect short-lived protein–protein or protein–DNA interactions and determine precise relative stoichiometries.

Here, we describe SM biochemical assays for CMG formation and helicase activation that recapitulate previously demonstrated dependence on Sld3/7 and DDK. Initial recruitment of Cdc45 to the Mcm2-7 double hexamer occurs in a stepwise fashion. We observe multiple Cdc45 and GINS proteins binding to individual Mcm2-7 double hexamers. Importantly, DDK levels modulate the number of these binding events and the frequency of final CMG formation. Consistent with these initial binding events being intermediates formed on the DDK-phosphorylated Mcm2-7 N-terminal tails, reducing the number of phosphorylation sites on the Mcm4 and Mcm6 N-terminal tail decreases the number of these intermediates and the efficiency of CMG formation. Importantly, the same phospho-site mutations in the Mcm4 or Mcm6 N-terminal tails reduce replication initiation frequency in vivo. Our findings support a model in which helicase activation is controlled by a combination of DDK-dependent regulation of the multiplicity of initial complexes between Cdc45, GINS, and the Mcm2-7 tails (the CtG complex) and the inefficient conversion of these CtGs to CMG complexes.

## Results

### An SM assay for CMG formation and DNA unwinding

We used colocalization single-molecule spectroscopy (CoSMoS, *Friedman et al., 2006*; *Hoskins et al., 2011*) to observe the process of eukaryotic CMG formation. This SM approach monitors the colocalization of fluorescently labeled proteins with individual surface-tethered fluorescently labeled DNA molecules to measure protein–DNA binding events. Fluorescent labeling of Mcm2-7, Cdc45, and GINS was accomplished using a SNAP-tag (Mcm2-7, *Gendreizig et al., 2003*; *Ticau et al., 2015*) or sortase-mediated coupling of fluorescent peptides (GINS, Cdc45, and Mcm2-7; *Popp et al., 2007*; *Ticau et al., 2015*, see Materials and methods). These fluorescent tags

did not interfere with protein function, as assessed by ensemble CMG-formation assays (*Figure 1—figure supplement 2*). To prevent loaded Mcm2-7 complexes from sliding off the template prior to CMG formation (*Evrin et al., 2009*; *Remus and Diffley, 2009*), we used a 1.2-kb circular DNA containing a single origin of replication. Comparison of the circular template with a 1.3-kb linear template (*Ticau et al., 2015*) showed that the circular template exhibited a twofold higher occupancy by Mcm2-7$^{4SNAP549}$ at the end of a 20-min SM helicase-loading assay (*Figure 1—figure supplement 3*).

To perform the CMG-formation reaction in an SM format, we executed a series of sequential incubations and monitored protein fluorescence (*Figure 1A*). After determining the locations of surface-coupled DNA molecules, we sequentially loaded the Mcm2-7$^{4SNAP549}$ helicase, phosphorylated the loaded helicases with DDK, and then added the remaining proteins necessary to form the CMG (*Figure 1—figure supplement 1*, S-CDK, Sld3/7, Cdc45, Dpb11, Sld2, GINS, Pol ε; *Yeeles et al., 2015*; *Heller et al., 2011*; *Kanemaki and Labib, 2006*; *Kamimura et al., 2001*).

To ensure that only loaded Mcm2-7 molecules were present during CMG formation, we included a high-salt wash (HSW1) after DDK phosphorylation to remove helicase-loading intermediates. We initially focused on the association of the Cdc45 component of CMG by labeling this protein with a distinct fluorophore (Cdc45$^{SORT649}$). To focus our observations on the events of CMG formation after helicase loading, we only monitored protein fluorescence after DDK phosphorylation of loaded Mcm2-7$^{4SNAP549}$ double hexamers (*Figure 1A*). During this period, each DNA molecule was continuously monitored for fluorescent-protein colocalization for ~30 min. After the CMG-formation reaction, we performed a second high-salt wash (HSW2) to distinguish fully formed, salt-resistant CMG complexes from intermediates that are readily removed by this treatment (*Yeeles et al., 2015*).

Analysis of the DNA-associated proteins after the experiment showed multiple hallmarks of CMG formation. Consistent with efficient CMG formation, more than half (0.53 ± 0.04) of the DNAs with loaded Mcm2-7$^{4SNAP549}$ complexes were bound to Cdc45$^{SORT649}$ when observed immediately after the HSW2 high-salt wash (*Figure 1B*). These binding events were eliminated in the absence of Sld3/7, consistent with Cdc45$^{SORT649}$ recruitment to Mcm2-7 relying on Sld3/7 (*Figure 1B*). Similarly, when Cdc6 was eliminated from the first step of the reaction to prevent helicase loading, subsequent CMG formation was dramatically reduced (fraction of DNAs bound by Cdc45 was 0.12 ± 0.01 with Cdc6 vs. 0.006 ± 0.006 without, *Figure 1—figure supplement 4*). Thus, CMG formation was dependent on prior helicase loading. In the absence of DDK, high-salt-resistant bound Cdc45 was reduced but not eliminated (*Figure 1B*). We hypothesized that these 'DDK-independent' binding events were due to residual DDK phosphorylation present on Mcm2-7 complexes purified from G1-arrested cells (*Heller et al., 2011*; *Tanaka et al., 2011*). Consistent with this idea, phosphatase treatment of Mcm2-7$^{4SNAP549}$ prior to helicase loading eliminated CMG formation in the absence of DDK (*Figure 1B*). We confirmed that phosphatase-treated Mcm2-7$^{4SNAP549}$ remained functional by sequential treatments with phosphatase and DDK. This sequence of treatments rescued CMG formation both in the SM (*Figure 1B*) and ensemble (*Figure 1—figure supplement 5*) settings, demonstrating that the observed CMG formation was fully DDK-dependent.

To assess the functionality of the CMGs assembled in the SM setting, we asked whether they were able to unwind DNA. To this end, we generated a linear 1.1-kb origin-containing DNA template with a fluorescent dye and a non-fluorescent quencher (BHQ-2) attached to opposing DNA strands at the surface-distal end of the DNA (*Kose et al., 2019*; *Figure 1C*). DNA unwinding will promote separation of the fluorescent dye-attached strand from the quencher-attached strand, leading to increased dye emission. We simultaneously monitored labeled Mcm2-7$^{4SNAP549}$ and unquenching of the DNA-associated fluorophore in the presence of the proteins required for CMG formation and two additional proteins (Mcm10 and replication protein A [RPA]) required for extensive DNA unwinding (*Douglas et al., 2018*). Consistent with previous in vitro assays that showed only a subset of loaded Mcm2-7 are activated (*Douglas et al., 2018*), we observed increased fluorescence for a fraction of DNAs with bound Mcm2-7$^{4SNAP549}$ (0.24 ± 0.04, *Figure 1D*). Importantly, elimination of helicase-activating factors (S-CDK, Cdc45, GINS, Dpb11, Pol ε, Sld3/7, RPA, and Mcm10) from the reaction prevented all unquenching events (*Figure 1D*). Together, these studies demonstrate that our SM assays for CMG formation and helicase activation exhibit the outcomes and protein dependencies expected based on previous in vivo and ensemble in vitro studies (*Kanemaki and Labib, 2006*; *Ilves et al., 2010*; *Heller et al., 2011*; *Tanaka et al., 2011*; *Yeeles et al., 2015*).

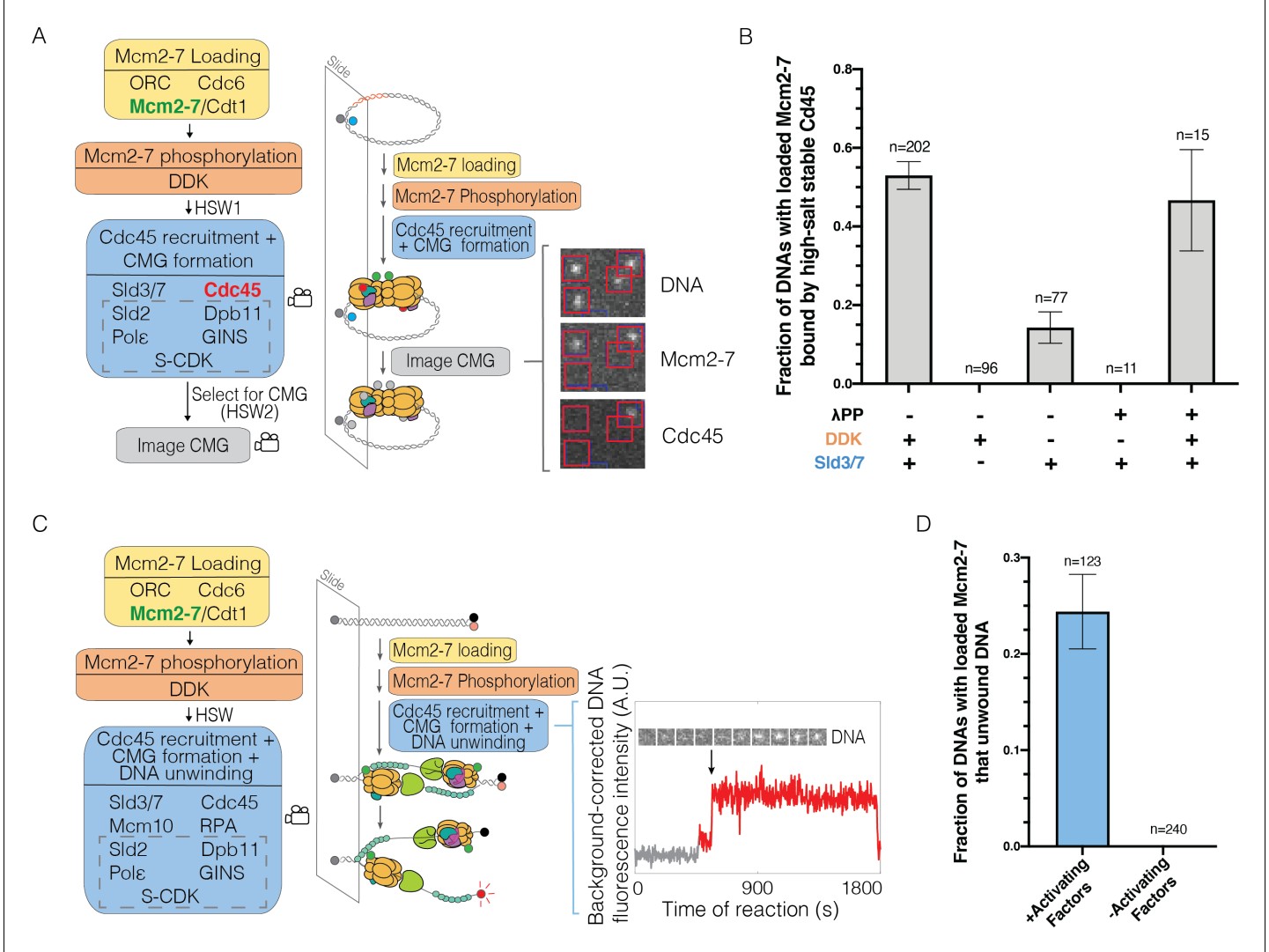

**Figure 1.** Single-molecule reaction for Cdc45-Mcm2-7-GINS (CMG) formation and DNA unwinding. (**A**) Schematic for the single-molecule CMG formation reaction. Alexa-Fluor-488-labeled (blue circle) circular origin DNA molecules were tethered to the slide surface. Purified Mcm2-7[4SNAP549] (green circle), Cdc45[SORT649] (red circle), and other indicated proteins necessary to form the CMG were incubated with slide-coupled DNA in the steps shown. Members of the group of proteins referred to as SDPGC are in the dashed box. A high-salt wash (HSW1) was performed after Mcm2-7 phosphorylation to remove helicase-loading intermediates. Colocalization of fluorescently labeled proteins with fluorescently labeled DNA was monitored (indicated by camera icon) during CMG formation and after a second high-salt wash (HSW2). Example images show a small subregion of the microscope field of view taken at a single time point after the CMG formation reaction recorded in the color channels for DNA[488], Mcm2-7[4SNAP549], and Cdc45[SORT649]. Red squares are centered on DNA locations. Spots within the red square indicate stable binding of Mcm2-7[4SNAP549] or Cdc45[SORT649] after HSW2. (**B**) Cdc45 binding depends on Sld3/7 and DDK phosphorylation. The fraction (± SE) of n DNA molecules with bound Mcm2-7[4SNAP549] that exhibited high-salt-resistant Cdc45[SORT649] at the first time point after the HSW2 is plotted for each of the conditions. DNAs are counted if they contain Mcm2-7[4SNAP549] at the beginning of the CMG formation reaction. Where indicated, lambda phosphatase (λPP) was used to dephosphorylate Mcm2-7[4SNAP549] prior to the Mcm2-7 loading reaction. The first bar represents the results of three replicates, the second and third bars represent the results of two replicates, and the remaining bars are the result of a single experiment. (**C**) Schematic of the single-molecule DNA-unwinding reaction. Cy5- and BHQ-2-labeled (red and black circles) linear origin-DNA molecules were coupled to microscope slides. The same stepwise incubations as with the single-molecule CMG-formation assay were used, except Mcm10 and replication protein A (RPA) were added to the 'CMG formation and DNA unwinding step'. The plot displays a representative background-corrected fluorescence-intensity record (see Materials and methods) for DNA[Cy5] (increase in fluorescence indicated by arrow). An objective image-analysis algorithm (***Friedman and Gelles, 2015***) detected a spot of DNA fluorescence at time points shown in red. (**D**) DNA unwinding is dependent on activating factors. Helicase-activating factors eliminated were S-CDK, Sld2, Dpb11, Pol ε, GINS, Sld3/7, Cdc45, Mcm10, and RPA. The fraction (± SE) of n DNA molecules that were unwound is plotted for both conditions. The online version of this article includes the following figure supplement(s) for figure 1:

**Figure supplement 1.** Schematic representation of DNA replication initiation.

**Figure supplement 2.** Fluorescently labeled proteins function at wild-type levels in ensemble Cdc45-Mcm2-7-GINS (CMG)-formation assays.

*Figure 1 continued on next page*

*Figure 1 continued*

**Figure supplement 3.** Circular DNA results in a higher number of retained Mcm2-7 during helicase loading.
**Figure supplement 4.** Cdc45 binds DNA in a Mcm2-7-loading-dependent manner.
**Figure supplement 5.** Phosphatase-treated Mcm2-7 remains functional after treatment with phosphatase.

## Cdc45 binding events to loaded Mcm2-7 are sequential

We next investigated the protein requirements for recruitment of Cdc45$^{SORT649}$ to the Mcm2-7$^{4SNAP549}$-DNA complex. Initially, we tested a minimal set of proteins predicted to recruit Cdc45 to loaded Mcm2-7 (*Kanemaki and Labib, 2006*; *Heller et al., 2011*; *Tanaka et al., 2011*). In these experiments, after phosphorylation of loaded Mcm2-7$^{4SNAP549}$ with 1.3 nM DDK, we added only Sld3/7 and Cdc45$^{SORT649}$ (*Figure 2A*). Under these conditions, we detected dynamic binding of one or two Cdc45$^{SORT649}$ with the majority of Mcm2-7$^{4SNAP549}$-bound DNA molecules (136/198, *Figure 2A*, *Figure 2—figure supplement 1*). A subset (62/198) of DNAs with loaded Mcm2-7$^{4SNAP549}$ never showed bound Cdc45. Consistent with the dynamic nature of the observed Cdc45 interactions and the requirement for additional proteins for CMG formation, in a separate experiment we did not detect high-salt-resistant Cdc45 binding to Mcm2-7 (0/70).

To assess the number of Cdc45 molecules bound to loaded Mcm2-7 more accurately, we constructed histograms that compiled Cdc45$^{SORT649}$ intensities measured at all Mcm2-7$^{4SNAP549}$-bound DNAs in every frame of the recorded images. We modeled these histogram data as the sum of equally spaced Gaussian peaks representing molecules with zero, one, two, or more bound Cdc45$^{SORT649}$. The model assumed independent binding of Cdc45 to multiple identical sites on Mcm2-7 (see Materials and methods). Based on this modeling, we determined the fraction of time a specific number of Cdc45$^{SORT649}$ proteins were bound to DNAs with loaded Mcm2-7$^{4SNAP549}$ (*Figure 2A, Ci*). This analysis showed that one Cdc45$^{SORT649}$ was bound to a DNA with an Mcm2-7$^{4SNAP549}$ double hexamer $0.222 \pm 0.002$ of the time and two Cdc45$^{SORT649}$ were bound only $0.079 \pm 0.001$ of the time (*Supplementary file 1a*). We also used this analysis to determine the average number of Cdc45 proteins bound to Mcm2-7$^{4SNAP549}$. Taking into consideration our finding that $0.73 \pm 0.07$ of the Cdc45 protein was labeled (see Materials and methods), we found that, on average, less than one Cdc45 molecule (labeled plus unlabeled) was associated with a loaded Mcm2-7 double hexamer ($0.60 \pm 0.06$, *Figure 2Ci*).

We considered the possibility that the stepwise increases in Cdc45$^{SORT649}$ fluorescence intensity (e.g., *Figure 2A*) represented transitions between states with two and four Cdc45 molecules instead of states with one and two molecules. If this were the case, however, we would expect to see a large subset of the bound Cdc45 to show half the fluorescent intensity relative to other events on the same DNA molecule. The reason for this expectation is that only a $0.73 \pm 0.07$ fraction of the Cdc45 protein is fluorescently labeled. Given this percentage of protein labeling, if the changes in fluorescence intensity were consistently due to simultaneous binding of two Cdc45 molecules, we would expect one of those molecules to be unlabeled $2P(1–P) = 0.40$ of the time, where $P$ is the fraction labeled. Given the multiplicity of Cdc45 binding events observed on most DNAs, such a situation would result in two levels of increases in fluorescent intensity in any given trace. Instead, we observed Cdc45$^{SORT649}$ fluorescence-intensity changes that were consistently similar across a given trace (*Figure 2A*, bottom panel, *Figure 2—figure supplement 1*). There are some events that might represent dimers of Cdc45 binding (e.g., *Figure 2A*, bottom, ~900 s), but these are rare and likely to be caused by two individual Cdc45$^{SORT649}$ molecules binding in rapid succession. Thus, we conclude that the multiple Cdc45s binding to Mcm2-7-bound DNA are recruited as monomers in a sequential fashion.

We then assessed how Cdc45$^{SORT649}$ binding events changed with the addition of the Sld2, Dpb11, Pol ε, GINS, and S-CDK (SDPGC) proteins, all of which are required for CMG formation (*Figure 2B*, *Figure 2—figure supplement 2*). We added these proteins to an otherwise unchanged reaction using 1.3 nM DDK. Fluorescent analysis showed that the addition of the SDPGC proteins lowered the fraction of time that DNA molecules bound by Mcm2-7$^{4SNAP549}$ were not associated with Cdc45$^{SORT649}$ (− SDPGC $0.682 \pm 0.001$; + SDPGC, $0.560 \pm 0.001$, *Supplementary file 1a*, rows 1 and 2). Consistent with this change, the average number of Cdc45 proteins (labeled plus unlabeled) on a Mcm2-7$^{4SNAP549}$-bound DNA increased in these conditions (− SDPGC $0.60 \pm 0.06$;

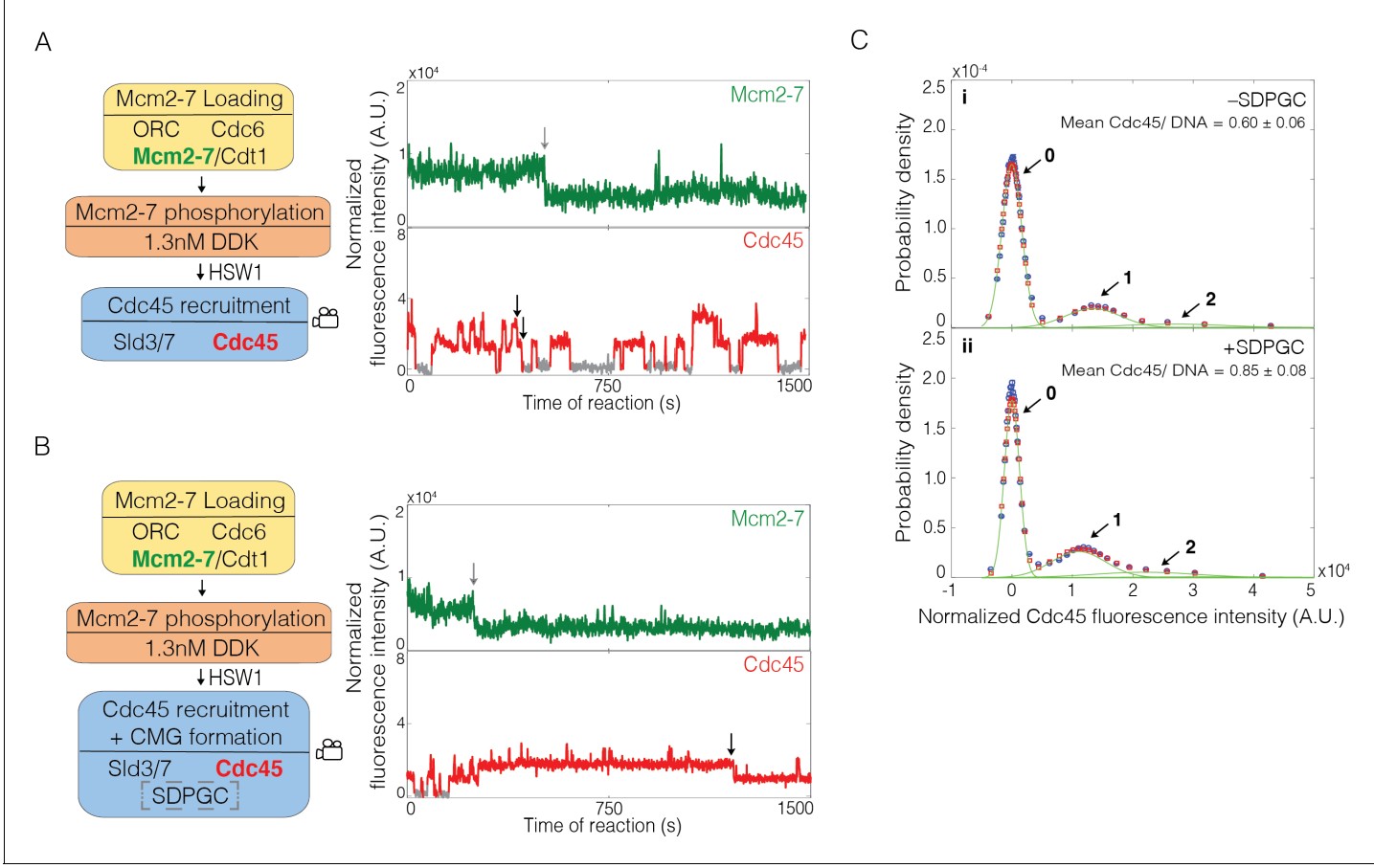

**Figure 2.** Sld2, Dpb11, Pol ε, GINS, and S-CDK (SDPGC) increases the frequency of Cdc45 binding events to the Mcm2-7 double hexamer. (**A**) Experiment protocol and representative fluorescence-intensity record for Mcm2-7[4SNAP549] and Cdc45[SORT649] at an origin-DNA location with minimal set of proteins required to recruit Cdc45. An objective image-analysis algorithm (*Friedman and Gelles, 2015*) detected a spot of protein fluorescence at time points shown in red or green. Photobleaching of Mcm2-7[4SNAP549] is marked by a gray arrow, and a set of likely dissociations of Cdc45[SORT649] molecules are marked by black arrows. (**B**) Experiment protocol and representative fluorescence-intensity record for Mcm2-7[4SNAP549] and Cdc45[SORT649] at an origin-DNA location with all the factors required for Cdc45-Mcm2-7-GINS (CMG) formation (+ SDPGC = Sld2, Dpb11, Pol ε, GINS, and S-CDK). Photobleaching of Mcm2-7[4SNAP549] is marked by a gray arrow, and a potential dissociation of Cdc45[SORT649] molecules is marked by a black arrow. (**C**) Multiple Cdc45[SORT649] molecules bind to Mcm2–7[4SNAP549]-bound DNA. Cdc45[SORT649] fluorescence-intensity histograms (blue) are shown for two conditions: – SDPGC (i) and + SDPGC (ii). The histogram data were fit to a sum-of-Gaussians model (red; see Materials and methods). Fit parameters and calculated area fractions of individual Gaussian components (green) corresponding to the presence of the indicated numbers of Cdc45[SORT649] molecules are given in *Supplementary file 1a*. The mean (±SE) numbers of Cdc45 molecules per DNA are calculated from the fit parameters and the fraction Cdc45 labeled. In all three panels, intensity values were background corrected and normalized as described in Materials and methods. The online version of this article includes the following figure supplement(s) for figure 2:

**Figure supplement 1.** Representative records of Cdc45[SORT649] fluorescence intensity.

**Figure supplement 2.** Representative records of Cdc45[SORT649] fluorescence intensity.

+ SDPGC, 0.85 ± 0.08, *Figure 2Ci, ii*). The addition of the SDPGC proteins did not alter the uniformity of Cdc45[SORT649] fluorescence-intensity increases, consistent with Cdc45 still being recruited as a monomer in a sequential fashion (*Figure 2B*, *Figure 2—figure supplement 2*). In contrast to SM experiments lacking the SDPGC proteins that showed no high-salt-resistant Cdc45 binding (0/70), the presence of the SDPGC proteins results in a fraction of loaded Mcm2-7 complexes forming the high-salt-resistant Cdc45 complexes characteristic of CMG formation (0.15 ± 0.05, *Figure 3A*). Together, these results show that the complete set of proteins required for CMG formation leads to additional Cdc45 binding to loaded Mcm2-7, a subset of which go on to form CMG complexes.

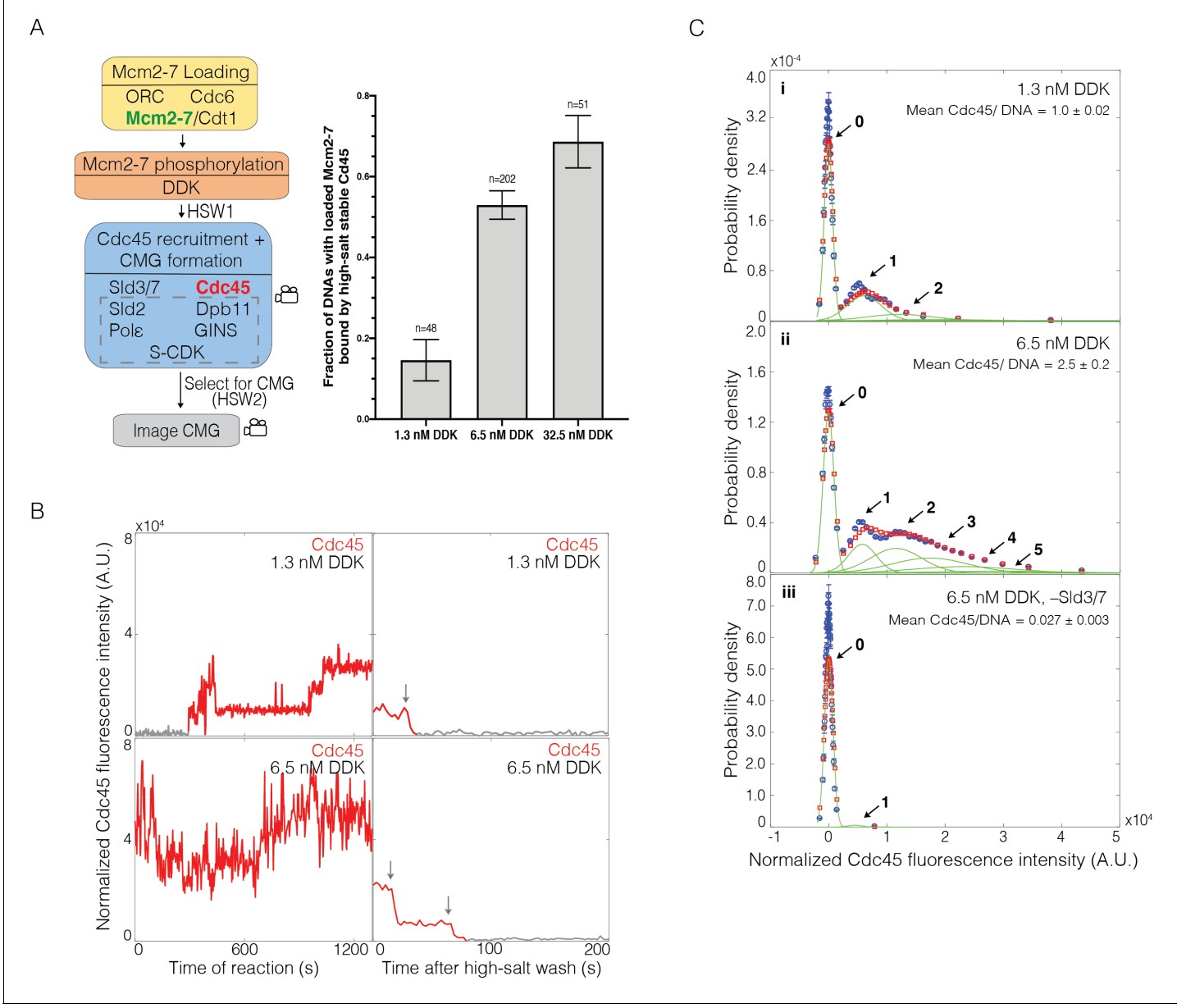

**Figure 3.** Dbf4-dependent kinase (DDK) modulates the number of Cdc45 binding events and Cdc45-Mcm2-7-GINS (CMG) formation efficiency. (**A**) Increasing DDK concentration results in more efficient CMG formation. The fraction (± SE) of n DNA molecules with loaded Mcm2-7[4SNAP549] that were bound by high-salt-stable Cdc45[SORT649] are reported for reactions using 1.3 nM, 6.5 nM, or 32.5 nM DDK. Experimental protocol was as in *Figure 1A*. The data from 6.5 nM DDK condition is the same as *Figure 1B*, left bar. (**B**) Representative fluorescence-intensity records for Cdc45[SORT649] under 1.3 nM and 6.5 nM DDK concentrations. Left panel is during CMG formation and right panel is after high-salt wash (HSW2). Points colored as in *Figure 2B*. Photobleaching of Cdc45[SORT649] after the HSW2 is marked by gray arrows. Fluorescence intensities after HSW2 were adjusted to account for the higher excitation laser intensity used during photobleaching. (**C**) Multiple Cdc45[SORT649] molecules bind to Mcm2-7[4SNAP549]. Cdc45[SORT649] fluorescence-intensity histograms (blue) are shown for three conditions: (i) 1.3 nM DDK; (ii) 6.5 nM DDK; and (iii) 6.5nM DDK in the absence of Sld3/7. The histogram data were fit to a sum-of-Gaussians model (red; see Materials and methods). Fit parameters and calculated area fractions of individual Gaussian components (green) corresponding to the presence of the indicated numbers of Cdc45[SORT649] molecules are given in *Supplementary file 1a*. The mean (± SE) numbers of Cdc45 molecules per DNA are calculated from the fit parameters and the fraction Cdc45 labeled.

The online version of this article includes the following figure supplement(s) for figure 3:

**Figure supplement 1.** Cdc45[SORT649] photobleaching events are temporally resolved.

**Figure supplement 2.** Cdc45-Mcm2-7-GINS (CMG) formation on each Mcm2-7 in the double hexamer can occur independently.

## DDK levels modulate the frequency of CMG formation

Our early experiments with all of the proteins required for CMG formation yielded a relatively low fraction of loaded Mcm2-7 complexes being converted to CMGs. These results led us to consider approaches to increase the efficiency of CMG formation. Because higher DDK levels improve origin initiation efficiency (*Mantiero et al., 2011*; *Tanaka et al., 2011*), we explored the impact of increasing DDK concentration. At the higher DDK concentration used in *Figure 1* (fivefold increase from 1.3 nM to 6.5 nM), the fraction of Mcm2-7$^{4SNAP549}$-bound DNAs that became bound by high-salt-resistant Cdc45$^{SORT649}$ increased from 0.15 ± 0.05 to 0.53 ± 0.04 (*Figure 3A*). Elevating DDK concentration fivefold further (to 32.5 nM) increased the fraction of loaded Mcm2-7 complexes forming CMGs from 0.53 ± 0.04 to 0.69 ± 0.06 (*Figure 3A*). Because the relative increase in CMG formation was more dramatic between 1.3 nM and 6.5 nM (suggesting we were approaching saturation at 32.5 nM), we focused our subsequent studies on these two DDK concentrations.

Increased DDK levels led to each Mcm2-7$^{4SNAP549}$-bound DNA interacting with more Cdc45$^{SORT649}$ proteins during the CMG formation reaction (*Figure 3B*, left panels). Fluorescence-intensity analysis revealed that elevating DDK levels to 6.5 nM increased the average number of Cdc45 molecules (labeled plus unlabeled) bound to Mcm2-7 by more than twofold (1.0 ± 0.1 vs. 2.5 ± 0.2; *Figure 3C*, compare panels i and ii). This analysis also revealed sometimes four or more Cdc45$^{SORT649}$ molecules bound to an individual Mcm2-7$^{4SNAP549}$ double hexamer at 6.5 nM DDK. Importantly, all of the Cdc45$^{SORT649}$ binding events remain Sld3/7-dependent, suggesting that the Cdc45$^{SORT649}$ binding detected requires indirect binding with the Mcm2-7 tails via Sld3/7 (*Figure 3Ciii*). The fraction of DNAs with Mcm2-7$^{4SNAP549}$ bound at least once by Cdc45 is similar between the 1.3 nM (40 out of 48 molecules, 0.83 ± 0.05) and 6.5 nM (71/87, 0.82 ± 0.06) DDK conditions, indicating that increasing DDK is primarily impacting the number of Cdc45 molecules bound per double hexamer rather than the fraction of double hexamers that are capable of interacting with Cdc45. Finally, we note that although these data give a sense of the multiplicity of Cdc45 proteins bound to Mcm2-7 at a specific time, these binding events are dynamic. Thus, the number of Cdc45 binding events to an Mcm2-7 double hexamer over the course of the reaction is higher.

The multiplicity of the bound Cdc45 suggests that binding occurs on the Mcm4 and Mcm6 tails rather than the core of the Mcm2-7 complex. The core of each Mcm2-7 molecule is thought to have only one site that directly binds Cdc45 (the site of Cdc45 association in the CMG, *Yuan et al., 2016*, *Goswami et al., 2018*). In contrast, there are many Sld3-binding sites on both the Mcm4 and Mcm6 N-terminal tails (*Deegan et al., 2016*), each of which could recruit Cdc45. Thus, the appearance of more than two bound Cdc45 molecules indicates that at least a subset (and perhaps most, see below) of the Cdc45 binding events observed under high DDK conditions reflect Cdc45 binding (via Sld3) to the phosphorylated N-terminal tails.

## Only a subset of the multiple Mcm2-7-bound Cdc45 proteins form CMGs

To determine the number of Cdc45$^{SORT649}$ proteins retained after the second high-salt wash (HSW2, *Figure 1A*), we increased the laser intensity and monitored protein fluorescence until all the protein-bound dyes were photobleached. Because photobleaching is stochastic, each photobleaching event indicates the presence of one fluorescently labeled protein. Although our intensity model suggests that a given Mcm2-7 is capable of simultaneously binding four or more Cdc45 molecules during the CMG formation reaction (*Figure 3Cii*), we never (0 of 202 examined) observed that more than two Cdc45$^{SORT649}$ proteins remained bound to a Mcm2-7-bound DNA after the HSW2 high-salt wash (e. g., *Figure 3B*, right panel). In principle, it was possible that we undercounted photobleaching events that were not temporally resolved; however, comparison of the rate of photobleaching under these conditions to the time resolution of the experiment indicated that this would be rare (*Figure 3—figure supplement 1*). The maximum of two high-salt-resistant Cdc45s is consistent with the interpretation that high-salt resistance indicates that the Cdc45s are incorporated into CMGs (*Yeeles et al., 2015*). Thus, although increased DDK leads to more Cdc45 proteins binding simultaneously to each Mcm2-7-bound DNA, at most two are incorporated into the final CMG complexes.

We frequently observed only one high-salt-stable Cdc45$^{SORT649}$ associated with Mcm2-7-bound DNA, leading us to ask whether CMG formation occurred independently or was coordinated for the two Mcm2-7 complexes present in a loaded double hexamer. If CMG formation was coordinated for

the two Mcm2-7 complexes, we would expect most of the high-salt-resistant Cdc45 to occur in pairs. Using the measured fraction labeled of Cdc45$^{SORT649}$ (p=0.73 ± 0.07), we calculated that if CMG formation was coordinated, 0.57 ± 0.09 ($P^2$/{$P^2$ + 2$P$(1-$P$)}) of the bound high-salt-resistant Cdc45$^{SORT649}$ should include two labeled Cdc45s (*Figure 3—figure supplement 2*). In contrast, we observed that only 42/102 (0.41 ± 0.05) of the Mcm2-7-bound DNAs with salt-resistant Cdc45 included two labeled Cdc45s when CMG formation was performed with 6.5 nM DDK (*Figure 3—figure supplement 2*). When the same measurement is performed with 1.3 nM DDK, this fraction drops to 1/7 (0.14 ± 0.13), consistent with lower numbers of bound Cdc45 leading to more frequent formation of a single CMG within the Mcm2-7 double hexamer. These findings suggest that CMG formation on the two Mcm2-7 complexes in each double hexamer can occur independently.

## Cdc45 dwell times are similar at low and high DDK levels

We next asked whether the complex pattern of Cdc45 binding events observed at high DDK levels represented a fundamentally different type of Cdc45 association or an increased frequency of the same type of Cdc45 association observed at lower DDK levels. To distinguish between these two possibilities, we repeated the SM CMG-formation assay with Cdc45 having a reduced labeling fraction (0.036 ± 0.004, see Materials and methods). These conditions allowed us to observe binding of individual labeled Cdc45 molecules even in the high DDK context when many additional (unlabeled) Cdc45 were simultaneously bound to the same Mcm2-7-bound DNA (*Figure 4A*, *Figure 4—figure supplement 1*). Using these conditions, we determined the distributions of bound Cdc45 dwell times (*Figure 4B*). We did not see a significant difference in the dwell time distributions of Cdc45$^{SORT649}$ at 1.3 nM and 6.5 nM DDK, consistent with the hypothesis that high DDK levels increase the number of Cdc45 associations to Mcm2-7-bound DNA without changing their properties.

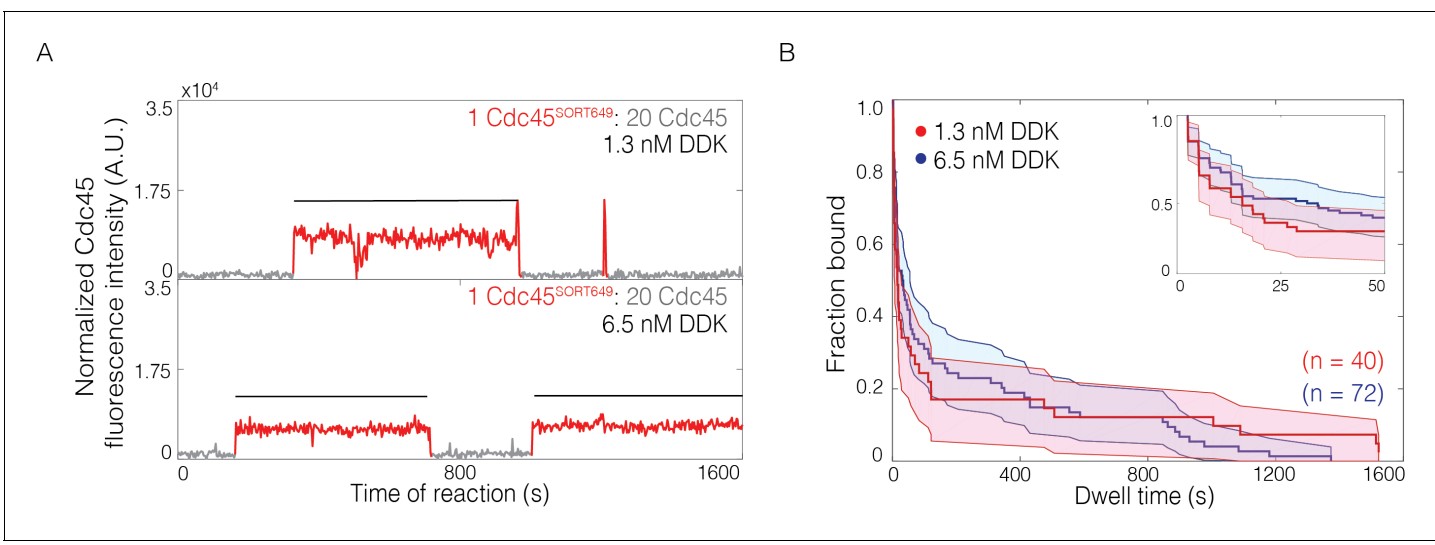

**Figure 4.** The lifetimes of individual bound Cdc45 molecules are not significantly changed by Dbf4-dependent kinase (DDK) levels. (**A**) Representative fluorescence-intensity records for Cdc45$^{SORT649}$ under conditions in which only a small fraction (0.036 ± 0.004) of Cdc45 molecules were fluorescently labeled at 1.3 nM (top) and 6.5 nM (bottom) DDK. Points colored as in *Figure 2B*. Black lines indicates single Cdc45$^{SORT649}$ binding. (**B**) Survival function for Cdc45$^{SORT649}$ dwell times on Mcm2-7-bound DNA. The vertical axis represents the fraction of Cdc45$^{SORT649}$ molecules that remain bound after the dwell interval indicated on the horizontal axis. Shaded areas represent the 95% confidence intervals for each curve. Inset: magnified view.

The online version of this article includes the following figure supplement(s) for figure 4:

**Figure supplement 1.** Reducing fraction of labeled Cdc45 results in the detection of individual binding events.

**Figure supplement 2.** Higher Dbf4-dependent kinase (DDK) concentrations result in higher numbers of phosphorylation events on Mcm6 and Mcm4.

## DDK levels modulate the number of Mcm2-7 tail phosphorylation events

Elevating DDK levels could change Mcm2-7 phosphorylation in multiple ways. More DDK could increase the number of phosphates per Mcm2-7 complex as well as the fraction of Mcm2-7 complexes that have any phosphates. DDK is known to exhibit stable binding to Mcm2-7 (*Sheu and Stillman, 2006*; *Ramer et al., 2013*; *Abd Wahab and Remus, 2020*), and an extreme model predicts that when DDK binds to Mcm2-7 this interaction leads to complete Mcm4- and Mcm6-tail phosphorylation. If this was the case, then DDK concentration would only modulate the fraction of Mcm2-7 complexes being completely phosphorylated, not the number of phosphates per Mcm2-7 tail. To investigate these possibilities, we exposed loaded Mcm2-7 complexes to increasing concentrations of DDK and monitored the extent of phosphorylation of Mcm6 using Phos-tag gels (*Figure 4—figure supplement 2A*). Changes in the mobility of Mcm6 indicated a distribution of numbers of phosphates, arguing against the latter model in which only complete tail phosphorylation occurs. Instead, as DDK concentration increases we see evidence for both increasing numbers of phosphates on the Mcm6 protein and a higher fraction of the molecules being modified. A similar experiment with the Mcm4 tail alone also showed elevated numbers of phosphorylation events when DDK concentration was increased (*Figure 4—figure supplement 2B*). These findings are consistent with DDK concentration regulating the number of Cdc45 binding through modulation of the number of Mcm4- and Mcm6-tail phosphorylation events.

## Cdc45-dependent binding of multiple GINS to Mcm2-7

To investigate the incorporation of GINS into CMGs, we performed an SM assay for CMG formation in which GINS was fluorescently labeled (*Figure 5A*). Several observations indicated that labeled GINS was being incorporated into CMGs. First, we observed Mcm2-7-bound DNAs interacting with high-salt-resistant GINS$^{SORT649}$ (*Figure 5A*). As we saw with Cdc45, the fraction of Mcm2-7-DNAs that bound to GINS$^{SORT649}$ increased when the concentration of DDK was raised from 1.3 to 6.5 nM (0.19 ± 0.04 vs. 0.72 ± 0.04; *Figure 5A*). Second, we never observed more than two high-salt-stable GINS$^{SORT649}$ after HSW2 (*Figure 5B*), and it is unlikely that we missed additional binding events due to simultaneous photobleaching of two GINS$^{SORT649}$ molecules (*Figure 5—figure supplement 1*). Third, elimination of proteins implicated in CMG formation interfered with high-salt-resistant GINS$^{SORT649}$ binding. Current models of CMG formation indicate that GINS recruitment requires S-CDK-phosphorylation-mediated interactions between Sld3, Dpb11, and Sld2. Consistent with this model, we observed a complete loss of high-salt-resistant GINS associations in the absence of Dpb11 (*Figure 5A*). Similarly, elimination of Cdc45 (0.09 ± 0.03) or Sld3/7 (0.07 ± 0.02) from the assay resulted in an eightfold to tenfold reduction in high-salt-resistant GINS$^{SORT649}$ bound to Mcm2-7$^{2SORT549}$-bound DNA (*Figure 5A*). Thus, we conclude that the high-salt-resistant GINS binding events detected are the result of CMG formation.

Investigation of the patterns of GINS binding during CMG formation supports the hypothesis that GINS is initially recruited to DDK-phosphorylated Mcm4 and Mcm6 tails. Because GINS is thought to be recruited to Mcm2-7 by indirect interactions (via Dpb11 and Sld2) with S-CDK-phosphorylated Sld3 (*Tanaka et al., 2007*; *Zegerman and Diffley, 2007*; *Muramatsu et al., 2010*), it was possible that we would observe the same DDK-phosphorylation-controlled multiplicity of GINS complexes binding to loaded Mcm2-7 that we observed for Cdc45. Alternatively, GINS recruitment could require interactions with the structured regions of Mcm2-7 involved in CMG formation. In that case, we would not observe more than two GINS bound. Consistent with the first model, increasing DDK levels led to the detection of elevated numbers of GINS$^{SORT649}$ binding to Mcm2-7$^{2SORT549}$-bound DNA molecules (*Figure 5B*). At 1.3 nM DDK, fluorescence-intensity analysis revealed primarily two or fewer GINS$^{SORT649}$ molecules binding to a loaded Mcm2-7$^{2SORT549}$ double hexamer (*Figure 5B, Ci*). Taking into account the fraction labeled of GINS (0.71 ± 0.07), this analysis indicated an average of 1.8 ± 0.2 total GINS molecules (labeled plus unlabeled) bound per Mcm2-7-bound DNA. In contrast, at 6.5 nM DDK, five or more GINS$^{SORT649}$ complexes could bind to an Mcm2-7$^{2SORT549}$-DNA complex with an average of 4.3 ± 0.4 GINS (labeled plus unlabeled) per Mcm2-7-bound DNA (*Figure 5Cii*). This average number of bound GINS$^{SORT649}$ is inconsistent with a model where all GINS interactions take place at the two GINS binding sites involved in CMG formation present in a double hexamer. As with Cdc45, the fraction of DNAs with Mcm2-7$^{2SORT549}$ that bound GINS$^{SORT649}$ at least once was similar for 1.3 nM (76/

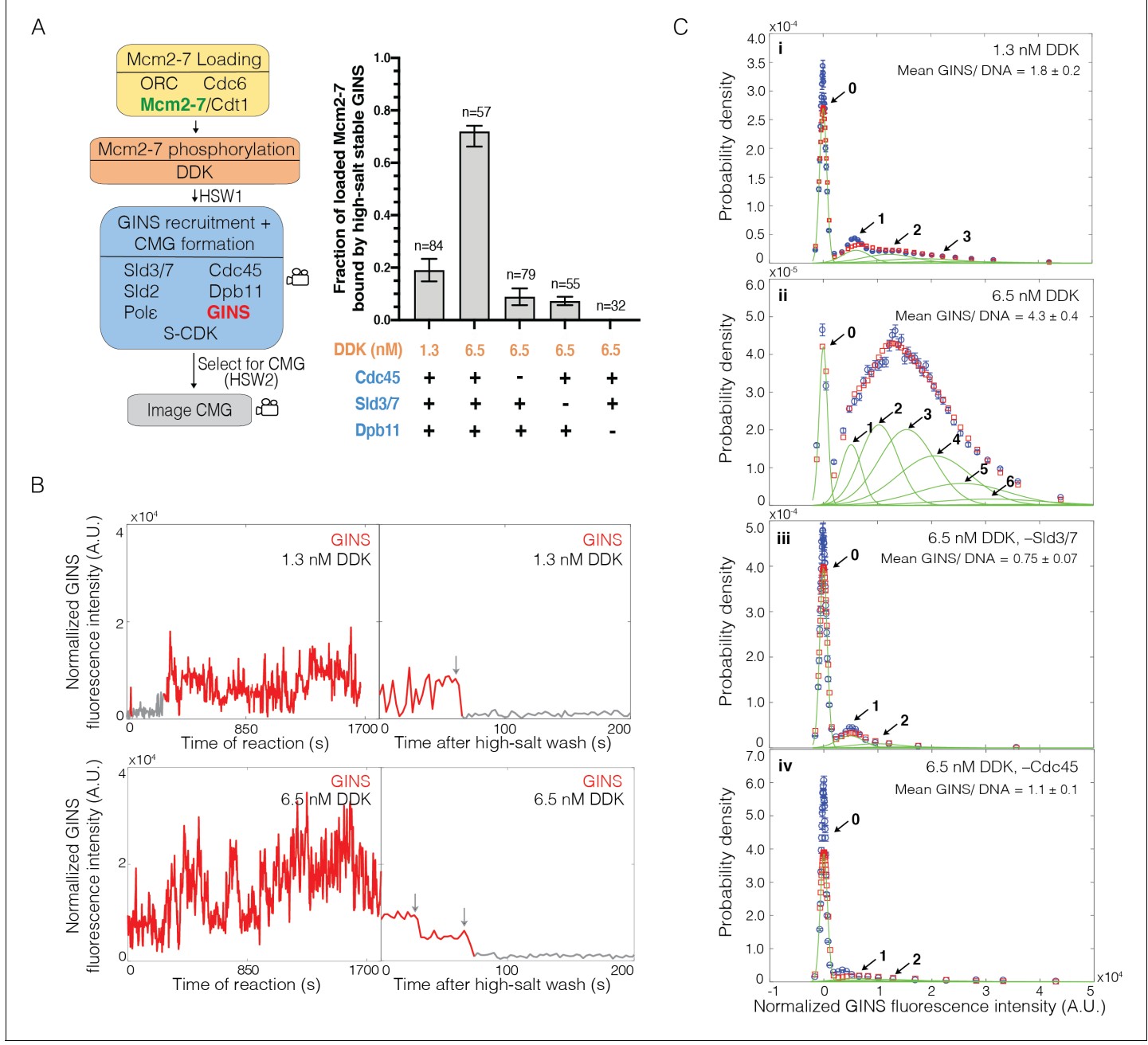

**Figure 5.** Multiple GINS bind to Mcm2-7. (**A**) Increasing Dbf4-dependent kinase (DDK) concentration results in more efficient Cdc45-Mcm2-7-GINS (CMG) formation that is dependent on the presence of Cdc45, Sld3/7, and Dpb11. Left: schematic representation of the experiment. Right: graph showing the fraction of n Mcm2-7^2SORT549^-bound DNA molecules that were bound by high-salt-stable GINS^SORT649^ under the indicated conditions. (**B**) Representative GINS^SORT649^ fluorescence-intensity records at two DNA molecules in the presence of 1.3 nM or 6.5 nM DDK. Left panel is during CMG formation and right panel is the same molecule after the second high-salt wash (HSW2). Points colored as in *Figure 2B*. Photobleaching of GINS^SORT649^ after HSW2 is marked by gray arrows. (**C**) GINS^SORT649^ fluorescence-intensity histograms (blue) are shown for four conditions: (i) 1.3 nM DDK; (ii) 6.5 nM DDK; (iii) 6.5 nM DDK in the absence of Sld3/7; and (iv) 6.5 nM DDK in the absence of Cdc45. The histogram data were fit to a sum-of-Gaussians model (red; see Materials and methods). Fit parameters and calculated area fractions of individual Gaussian components (green) corresponding to the presence of the indicated numbers of GINS^SORT649^ molecules are given in *Supplementary file 1b*. The mean (± SE) numbers of GINS molecules per DNA are calculated from the fit parameters and the fraction GINS labeled.

The online version of this article includes the following figure supplement(s) for figure 5:

**Figure supplement 1.** Photobleaching GINS^SORT649^ events are temporally resolved.

**Figure supplement 2.** Mcm2-7^2SORT549^ dependence of GINS^SORT649^ binding.

**Figure supplement 3.** Dpb11 dependence of GINS^SORT649^ binding.

84, 0.90 ± 0.04) and 6.5 nM (54/57, 0.95 ± 0.03). Because almost all complexes bound GINS under both conditions, the impact of DDK on CMG formation is due to increased numbers of GINS bound per Mcm2-7 double hexamer.

The ability of the SM CMG formation assay to detect association of GINS$^{SORT649}$ with Mcm2-7-bound DNA in real time allowed us to test the role of different proteins in initial recruitment of GINS. As expected, if these interactions occur only on Mcm2-7-bound DNA, we observed a dramatic reduction in GINS$^{SORT649}$ binding events for DNAs that were not associated with a labeled Mcm2-7 (*Figure 5—figure supplement 2*). Consistent with models indicating a protein interaction network between Sld3, Dpb11, Sld2 recruits GINS to Mcm2-7, we found that elimination of Dpb11 dramatically reduces the frequency and multiplicity of GINS$^{SORT649}$ binding to Mcm2-7$^{2SORT549}$ (*Figure 5—figure supplement 3*). We also asked if Cdc45 functioned during initial GINS association. It was possible that distinct Mcm2-7-bound Sld3 molecules recruit GINS and Cdc45, and they only come together later in CMG formation. Alternatively, GINS recruitment could require Sld3/7 to be bound to Cdc45. To distinguish between these models, we assessed GINS$^{SORT649}$ binding events during CMG formation in the absence of Cdc45 or Sld3/7. Elimination of either protein resulted in similar reductions in the numbers of Mcm2-7-bound GINS during CMG formation (*Figure 5C*, compare panels iii and iv). Elimination of either Cdc45 (1.1 ± 0.1) or Sld3/7 (0.7 ± 0.07) resulted in a fourfold reduction in the average number of GINS (labeled plus unlabeled) per Mcm2-7-bound DNA (*Figure 5Ciii, iv*). Thus, we conclude that the initial binding of GINS requires both Sld3/7 and Cdc45 and that the recruitment of GINS and Cdc45 occurs via the same Sld3/7 molecules. Importantly, we also see a consistent correlation between the multiplicity of GINS$^{SORT649}$ binding events (*Figure 5C*) and the frequency that these events are converted into CMGs (*Figure 5A*).

## Reducing DDK-dependent phosphorylation sites decreases CMG formation

Together, our data support a model in which complexes including both Cdc45 and GINS (and likely other proteins, see Discussion) form at multiple sites on the DDK-phosphorylated Mcm2-7 N-terminal tails. We refer to these complexes as Cdc45-tail-GINS (CtG) complexes. Our data also suggests that elevating the multiplicity of CtGs formation increases the probability of CMG formation. Based on these observations, we propose that the conversion of CtGs into CMG complexes is inefficient with multiple CtGs releasing prior to successful CMG formation.

To test the model that the multiple associations of Cdc45 involve interactions with DDK-phosphorylated Mcm4 and Mcm6, we purified and fluorescently labeled Mcm2-7 complexes with fewer DDK phosphorylation sites on either the Mcm6 (Mcm2-7$^{4SNAP549-6AD/E+ASP/Q}$) or Mcm4 (Mcm2-7$^{2SORT549-4AD/E+ASP/Q}$) N-terminal tails. These mutations ablated both direct DDK target sequences (S/T-D/E) and CDK- or Mec1-phosphorylation-dependent DDK target sequences (SS/T-P/Q) (*Masai et al., 2000*; *Cho et al., 2006*; *Mok et al., 2010*; *Randell et al., 2010*). The Mcm4 AD/E+ASP/Q mutant eliminated 14 sites and Mcm6 AD/E+ASP/Q eliminated 11 sites, and simultaneous elimination of both sets of sites results in cell death (*Randell et al., 2010*). We also purified and fluorescently labeled a Mcm2-7 complex that lacked the Mcm6 N-terminal tail (Mcm2-7$^{4SNAP549-6ΔN}$). Using each of these fluorescently labeled Mcm2-7 mutants and Cdc45$^{SORT649}$, we monitored the pattern of Cdc45$^{SORT649}$ binding and the extent of CMG formation using 6.5 nM DDK in the SM CMG formation assay.

Each of the Mcm2-7 mutants showed a significantly lower fraction of loaded Mcm2-7 molecules that were converted to CMGs when compared to WT Mcm2-7 (*Figure 6A*). In experiments with 6.5 nM DDK, Mcm4 (0.21 ± 0.07) and Mcm6 (0.30 ± 0.07) mutants reduced this fraction by approximately half relative to WT Mcm2-7, consistent with the elimination of DDK sites on one of the two tails. Interestingly, the impact of the Mcm6 tail deletion (0.04 ± 0.02) was substantially larger than that of the Mcm6 phosphorylation mutant. This observation suggests that either the deletion eliminates additional phosphorylation sites that impact CMG formation or that there are other functions beyond CtG formation performed by the Mcm6 N-terminal tail during helicase activation (see Discussion). Together, these data are consistent with the multiple DDK phosphorylation sites on the Mcm4 and Mcm6 N-terminal tails being an important element in the response of the helicase activation to DDK levels.

To ask if the reduced CMG formation observed for the Mcm2-7 mutants was due to lowering the multiplicity of CtG formation events, we monitored Cdc45$^{SORT649}$ association during CMG formation

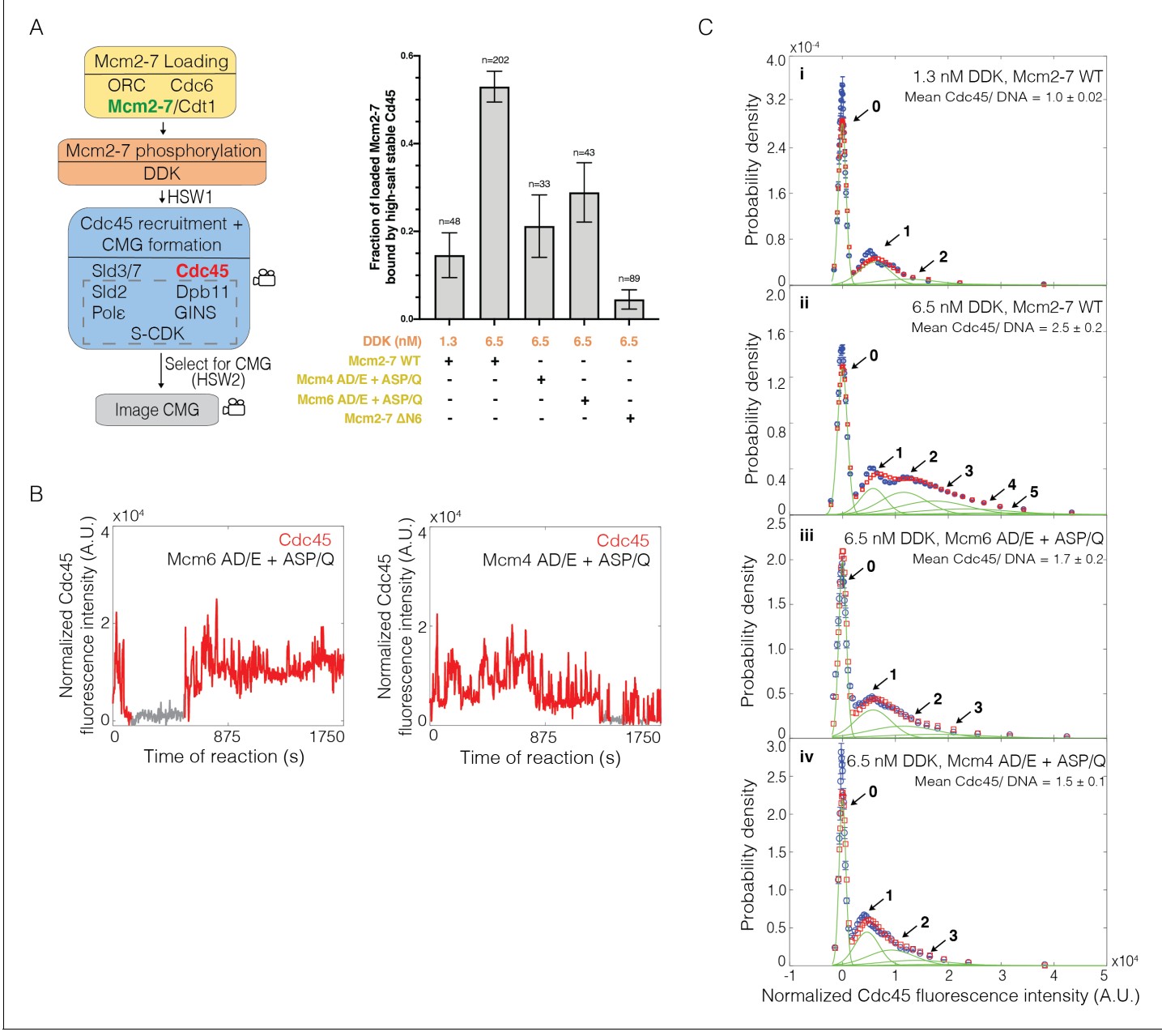

**Figure 6.** Loss of the Mcm6 N-terminal tail reduces Cdc45 binding events and Cdc45-Mcm2-7-GINS (CMG) formation. (**A**) Reduction of Dbf4-dependent kinase (DDK)-dependent phosphorylation sites on either Mcm6 or Mcm4 and deletion of the Mcm6 N-terminal tail reduced high-salt-stable Cdc45 binding to Mcm2-7-bound DNAs, indicative of CMG formation. The data from 1.3 nM and 6.5 nM DDK (first and second bars) are the same as *Figure 3A*. (**B**) Representative fluorescence-intensity records for Cdc45[SORT649] binding to Mcm2–7[4SNAP549–6AD/E+ASP/Q]- and Mcm2-7[2SORT549–4AD/E+ASP/Q]-bound DNA during CMG formation at 6.5 nM DDK. Points colored as in *Figure 2B*. (**C**) Binding of Cdc45[SORT649] to Mcm2–7[4SNAP549–6AD/E+ASP/Q]- and Mcm2-7[2SORT549–4AD/E+ASP/Q]-bound DNA. Cdc45[SORT649] fluorescence-intensity histograms (blue) are shown for four conditions: (i) 1.3 nM DDK with wild-type (WT) Mcm2-7[4SNAP549], (ii) 6.5 nM DDK with WT Mcm2-7[4SNAP549], (iii) 6.5 nM DDK with Mcm2-7[4SNAP549–6AD/E+ASP/Q], and (iv) 6.5 nM DDK with Mcm2-7[2SORT549–4AD/E+ASP/Q]. The data from 1.3 nM (i) and 6.5 nM (ii) DDK are the same as *Figure 3C*. Fit parameters and calculated area fractions of individual Gaussian components (green) corresponding to the presence of the indicated numbers of Cdc45[SORT649] molecules are given in *Supplementary file 1a*. The mean (± SE) number of Cdc45 molecules per DNA are calculated from the fit parameters and the fraction Cdc45 labeled.

The online version of this article includes the following figure supplement(s) for figure 6:

**Figure supplement 1.** Loss of the Mcm6 N-terminal tail reduces Cdc45 binding events.

at 6.5 nM DDK. We observed fewer Cdc45$^{SORT649}$ binding events, relative to wild-type Mcm2-7 at 6.5 nM DDK, for each of the Mcm6 or Mcm4 N-terminal mutations (compare *Figure 6B* and *Figure 6—figure supplement 1A* with *Figure 3B*, bottom panel). Fluorescence-intensity analysis revealed that the average number of Cdc45 (labeled plus unlabeled) bound to Mcm2-7$^{4SNAP549-6AD/E+ASP/Q}$, Mcm2-7$^{2SORT549-4AD/E+ASP/Q}$, or Mcm2-7$^{4SNAP549-6\Delta N}$ during CMG formation at 6.5 nM DDK was also lowered when compared to WT Mcm2-7 ($1.7 \pm 0.2$, $1.5 \pm 0.1$, and $0.10 \pm 0.01$, respectively; *Figure 6C*, *Figure 6—figure supplement 1B*). Thus, as with the protein elimination experiments (e. g., –Sld3/7, –Dpb11, and –Cdc45; *Figure 3* and *Figure 5*), we observe a correlation between the impact of the mutants on the multiplicity of CtG events with the efficiency of CMG formation (compare *Figure 6A and C*).

## Loss of Mcm4 or Mcm6 DDK-phosphorylation sites reduces origin firing

To test the hypothesis that the number of phosphorylation sites modulates CMG formation and helicase activation in vivo, we analyzed replication origin usage in WT cells and cells containing either the Mcm4 or Mcm6 phosphorylation-site-deficient mutant. To this end, G1 phase-synchronized cells were released into S phase with hydroxyurea (HU) and the nucleotide analog 5-bromo-2-deoxyuridine (BrdU). Because HU is an inhibitor of dNTP biosynthesis, only sequences including and adjacent to early activating origins of replication will incorporate BrdU (*Yabuki et al., 2002*). The reduced dNTP levels result in stalled forks that activate the intra-S-phase checkpoint to prevent initiation from late-firing origins of replication (*Iyer and Rhind, 2017*). The resulting DNA samples were immunoprecipitated with anti-BrdU and subjected to deep sequencing (*MacAlpine et al., 2004*).

The sequencing results show that the Mcm4 and Mcm6 DDK-phosphorylation-site mutants reduce but do not eliminate replication initiation. As expected, peaks of BrdU sequence reads coincide with locations of known early-initiating replication origins (*Figure 7A*). In this experiment, the normalized sequence read counts reflect the efficiency of early origin usage (*Yabuki et al., 2002*; *MacAlpine et al., 2004*). Importantly, we observe lower normalized read counts for the Mcm4 and Mcm6 mutant cells relative to wild-type, indicating that both mutants reduce early origin efficiency. We also measured the average read count for 5000 bp on either side of every yeast origin. As we saw for the individual origins, there is a strong reduction in initiation for both mutants in these data (*Figure 7B*). Together, these data indicate that the same mutations that alter CtG and CMG formation in vitro impact replication initiation in vivo, supporting a model in which modulating the ability of Mcm2-7 to form multiple CtG complexes in response to DDK levels alters origin initiation frequency in cells.

## Discussion

### Cdc45 and GINS are recruited by the Mcm2-7 N-terminal tails

The studies presented here provide evidence for a new intermediate in the helicase-activation process, the CtG, which can form at multiple sites on the Mcm2-7 N-terminal tails. We found that DDK levels modulate the number of these intermediates formed per loaded Mcm2-7 complex and that only a subset of CtGs transition to a CMG (*Figure 8*). We propose that the combination of these properties creates a mechanism that controls replication initiation efficiency by sensitizing helicase activation to DDK activity.

Previous studies found that Sld3 contains a phosphopeptide-binding domain that recognizes both DDK-dependent and DDK-independent phosphorylation sites on Mcm4 and Mcm6 N-terminal tails (*Deegan et al., 2016*). Binding of Sld3 to the DDK-dependent phosphorylation sites is required for the recruitment of Cdc45 to the Mcm2-7 double hexamer (*Deegan et al., 2016*). It was not clear, however, whether Sld3/7 recruited Cdc45 and GINS independently or as part of a single complex, how these interactions led to subsequent CMG formation, or how the many potential binding sites for Sld3/7 on Mcm4 and Mcm6 tails contributed.

Here, we present evidence that the initial recruitment of GINS and Cdc45 involves formation of a multi-protein complex with the Mcm2-7 N-terminal tails. First, there are only two non-tail binding sites on each Mcm2-7 double hexamer for Cdc45 or GINS (*Yuan et al., 2016*; *Goswami et al., 2018*), and yet we see more than two of these proteins bound during CMG formation (*Figure 3*, *Figure 5*). Second, binding of Cdc45 and GINS is strongly dependent on both DDK and Sld3/7

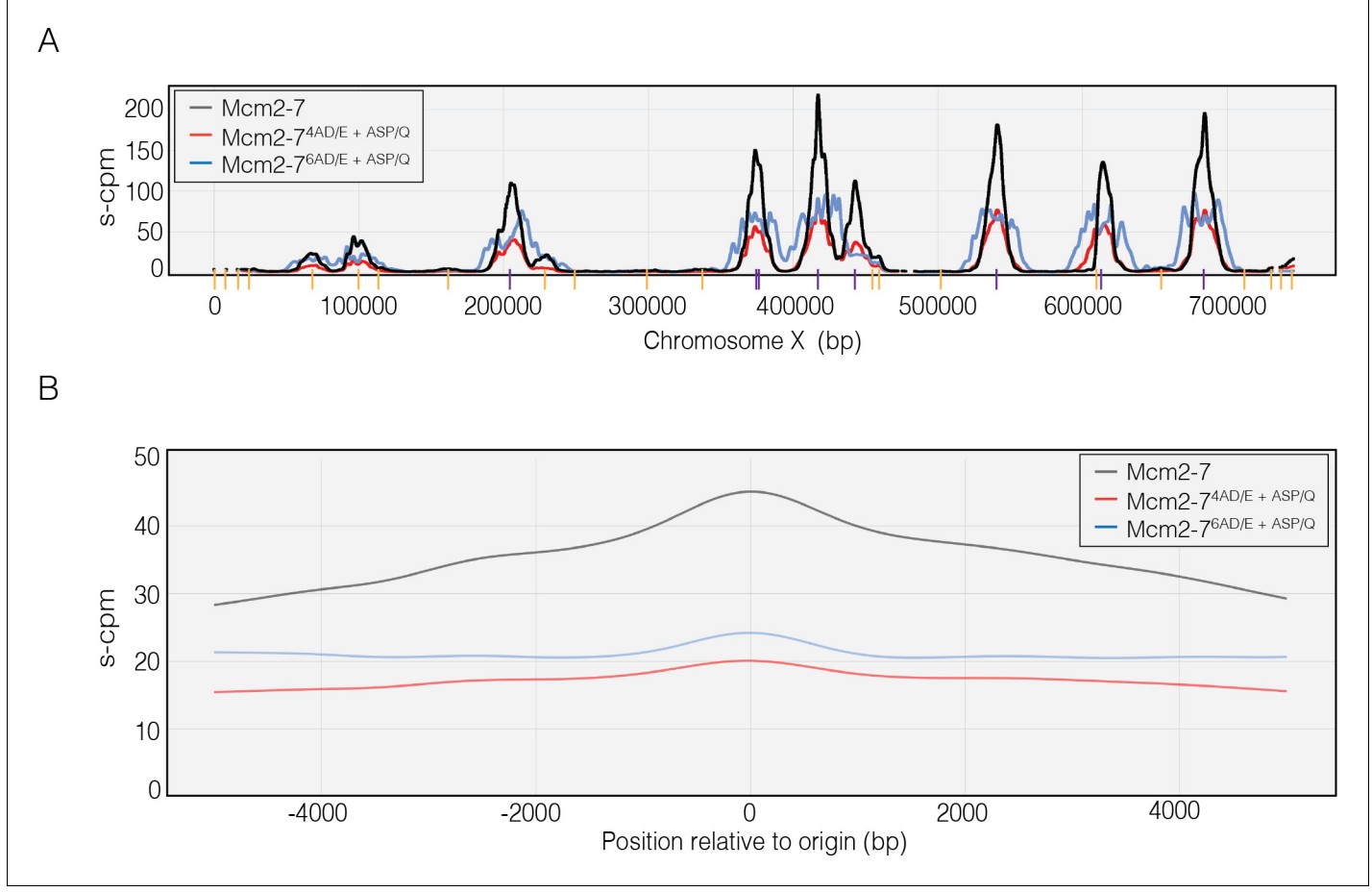

**Figure 7.** Dbf4-dependent kinase (DDK)-dependent phosphorylation sites on the N-terminal tails of Mcm4 and Mcm6 promote origin initiation. (**A**) Impact of Mcm4 and Mcm6 N-terminal DDK phosphosite mutations at specific origins of replication. Normalized sequence read numbers are reported for 5-bromo-2-deoxyuridine (BrdU)-labeled DNAs across the length of *Saccharomyces cerevisiae* chromosome X for strains expressing wild-type Mcm2–7 (yML514), Mcm2-7$^{4AD/E+ASP/Q}$ (yML512), or Mcm2-7$^{6AD/E+ASP/Q}$ (yML513). Known origins of replication are represented by vertical lines at the bottom of the plot. Early origins are represented with purple vertical lines (origins with time of replication <20 min; *Yabuki et al., 2002*; *Siow et al., 2012*), whereas non-early origins are represented by yellow vertical lines. (**B**) Impact of Mcm4 and Mcm6 N-terminal DDK phosphorylation site mutations across all origins. The average BrdU sequence read depth (s-cpm) for strains expressing wild-type Mcm2–7, Mcm2-7$^{4AD/E+ASP/Q}$, or Mcm2-7$^{6AD/E+ASP/Q}$ is reported for DNA sequences flanking all origins of replication. Origins were aligned with respect to the midpoint of each origin.

(*Figure 1B*, *Figure 3C*, *Figure 5C*). Because the Mcm4 and Mcm6 N-terminal tails are the targets of DDK and Sld3 binding (*Sheu and Stillman, 2006*; *Randell et al., 2010*; *Deegan et al., 2016*), this dependence is consistent with initial binding of Cdc45 and GINS at these sites. Third, the dwell time distributions of Cdc45 associations at high or low DDK levels are similar (*Figure 4*), arguing that the increased multiplicity of Cdc45 binding is not due to a distinct mechanism of Cdc45 binding at high DDK levels. Fourth, eliminating DDK-phosphorylation sites on either the Mcm4 or Mcm6 N-terminal tail greatly reduces CtG formation (*Figure 6*). Finally, the strong dependence of GINS recruitment on the presence of Cdc45 (*Figure 5C*) and the increased frequency of Cdc45 binding events in the presence of GINS and its associated factors (*Figure 2*) is consistent with a simple mechanism in which these proteins are recruited sequentially to the tails to form a single complex rather than binding to the tails at independent positions.

Our evidence also supports the CtG being an intermediate on the way to CMG formation. Consistent with this hypothesis, elimination of factors that promote CtG formation (e.g., DDK and Sld3/7) also inhibits CMG formation (*Figure 1B*, *Figure 5A*). Similarly, elimination of a subset of CtG binding sites by eliminating Mcm4 or Mcm6 DDK-phosphorylation or deleting the Mcm6 N-terminal tail results in a strong reduction in CMG formation that correlates with the effects on CtG formation (*Figure 6*). Indeed, throughout our studies we observe a consistent correlation between the

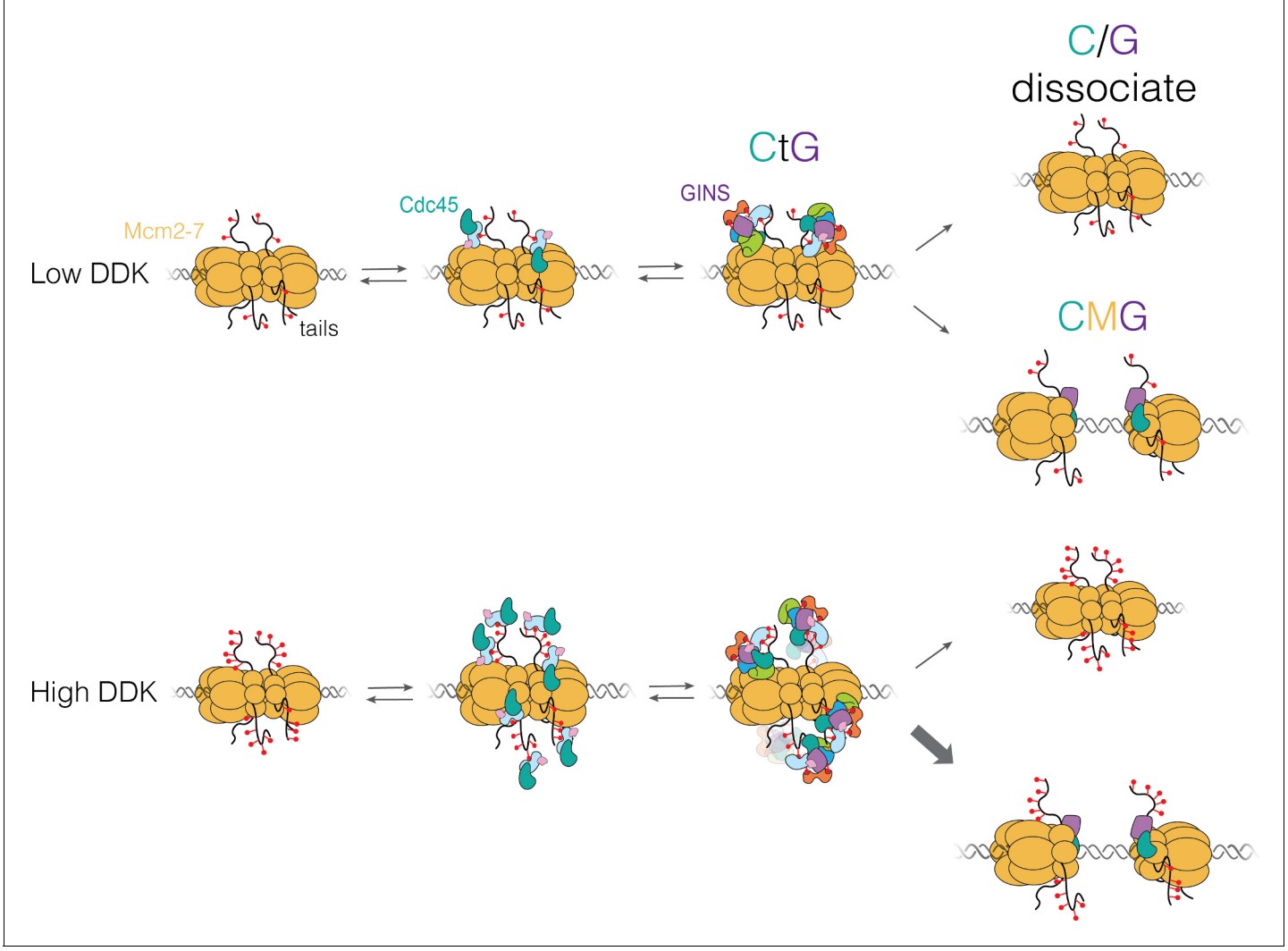

**Figure 8.** Proposed model for Cdc45-Mcm2-7-GINS (CMG) formation. Dbf4-dependent kinase (DDK) levels control the amount of Mcm4 and Mcm6 N-terminal tail phosphorylation. These modifications indirectly (via Sld3/7) recruit Cdc45 and GINS, forming the Cdc45-tail-GINS (CtG) intermediate. The CtG intermediate can then follow one of two paths: (i) dissociation or (ii) deposition onto the structured core of the Mcm2-7 helicase to form the CMG. We hypothesize that the rate of this conversion is the same for any CtG, but it is infrequent. Thus, having more CtGs (e.g., at higher DDK levels) increases the probability of CMG formation, which we assume is irreversible. Relevant proteins are labeled in the illustration; red lollipops represent phosphorylations. C/G: Cdc45-GINS.

multiplicity of CtG formation on a Mcm2-7 double hexamer and the efficiency of subsequent CMG formation.

We note that the impact of deleting the Mcm6 N-terminus on both CtG and CMG formation is larger than that observed for a mutation that merely eliminates DDK phosphorylation sites on Mcm6 (*Figure 6*, *Figure 6—figure supplement 1*). It is possible that there are additional non-DDK-dependent phosphorylation sites in Mcm6 that are contributing to CtG and CMG formation, and their elimination by the deletion further reduces the formation of these complexes (*Figure 6*, *Figure 6—figure supplement 1*). This model is consistent with the previous observation that not all Sld3 phosphopeptide-binding sites are canonical DDK-phosphorylation sites (*Deegan et al., 2016*). It is possible that such sites explain how cells survive without DDK in the context of certain Mcm2-7 mutations (*Sclafani et al., 2002*; *Sheu and Stillman, 2010*). It is also possible that forming CtGs on both the Mcm4 and Mcm6 N-terminal tails, rather than only on the Mcm4 tail, is advantageous for CMG formation. A second possibility is that there is an additional function of the Mcm6 tail that contributes to CMG formation. For example, the Mcm6 N-terminal tail could facilitate the CtG to CMG

transition. We note, however, that such functions cannot be essential since the Mcm6 N-terminal deletion is not lethal in vivo (*Champasa et al., 2019*).

## What are the components of the CtG?

Although our findings indicate that the CtG minimally includes Cdc45 and GINS bound to Mcm6 and Mcm4 N-terminal tails, this complex is likely to contain other proteins involved in CMG formation. The robust Sld3/7 dependence of Cdc45 and GINS binding to Mcm2-7 (*Figure 3*, *Figure 5*) suggests that the CtG includes this protein. A network of interactions between Sld3, Dpb11, Sld2, and GINS is required to form the CMG (*Tanaka et al., 2007*; *Zegerman and Diffley, 2007*; *Muramatsu et al., 2010*). Given Sld3's known binding to phosphorylated N-terminal tails (*Deegan et al., 2016*), the simplest model is that these interacting proteins are required to form CtGs, and subsequently CMGs. Consistent with this model, we observe nearly a complete loss of CtG formation in the absence of Dpb11 (*Figure 5C*). Finally, it is possible that Pol ε is simultaneously recruited with GINS to the CtG as part of a preformed pre-LC complex (*Muramatsu et al., 2010*).

The full set of roles performed by Sld3, Dpb11, Sld2, and Pol ε in CtG and CMG formation remains to be determined. These proteins could be involved only in CtG assembly or their release could facilitate the transition of Cdc45 and GINS binding with the Mcm2-7 tails to form the final interactions characteristic of the CMG. We note that in our experiments eliminating Sld3/7, Cdc45, or Dpb11, we see stronger effects of Dpb11 elimination than the other two proteins on GINS associations before and after a high-salt wash (*Figure 5*). One interesting explanation for this distinction is that Dpb11 (and perhaps Sld2 and Pol ε) allows GINS to interact with loaded Mcm2-7 at a low level independently of Cdc45 and Sld3/7. Given that Sld3/7 is the only protein involved in CMG formation known to interact with the Mcm4 and Mcm6 phosphorylated tails, this GINS interaction is more likely to be with the Mcm2-7 site involved in CMG formation. Consistent with this idea, we never observed more than two GINS binding events in the absence of Dpb11. Such a function would suggest a role for Dpb11 in establishing interactions with the core of the Mcm2-7 complex during the CtG to CMG transition. Finally, the requirement of Cdc45 to recruit GINS (*Figure 5Ciii*) strongly suggests that the CtG involves direct interactions between these proteins, suggesting the interesting possibility that key interactions between these proteins involved in the CMG are already present in the CtG.

## DDK effects are mediated by inefficient CtG to CMG conversion

Our studies indicate that the conversion from CtG to CMG is inefficient. With low DDK levels, we often observed two Cdc45 proteins that associated with Mcm2-7 double hexamers over long periods of time (>100 s; 33 long-lived binding events/48 double hexamers). However, only 7 of these 33 converted to CMGs (*Figure 3A*), indicating that most of the CtGs either dissociate or fail to become part of the CMG before the end of the experiment (*Figure 8*, top). We hypothesize that this inefficient conversion is true at all DDK concentrations. When DDK concentration was raised, we observed increased multiplicity of CtG formation (*Figure 3A*, *Figure 5A*). In contrast, increased DDK is not associated with a significant change in dwell times of individual Cdc45 molecules (*Figure 4*) or the fraction of Mcm2-7 double hexamers bound by Cdc45 or GINS. Based on these findings, we propose that, due to the inefficient conversion of CtGs to CMGs, the increased multiplicity of CtGs on a given double hexamer at higher DDK levels leads to increased opportunities for CtG conversion to CMG (*Figure 8*, bottom) and, therefore, increased rates of origin activation.

## Implications for the control of cellular origin activation

Our model is consistent with studies addressing the control of origin initiation efficiency and replication timing (*Mantiero et al., 2011*; *Tanaka et al., 2011*). These studies showed that increasing the expression of a subset of limiting helicase-activation proteins, including the Dbf4 regulatory subunit of DDK, increased origin initiation efficiency and advanced the time of replication initiation for late initiating origins. Our findings suggest that these changes would increase the number of CtGs formed per double hexamer and consequently the efficiency of CMG formation. We propose that the combination of multiple potential phosphorylation sites on the Mcm4 and Mcm6 tails, and the inefficient conversion of the CtGs formed on the tails to the CMG, tunes the helicase-activation process to respond to levels of DDK and other helicase-activation factors. Consistent with this model, reducing the number of phosphorylation sites on Mcm4 and Mcm6 or deleting the Mcm6 tail

reduced CtG and CMG formation at the same DDK levels (*Figure 6*). Importantly, we see that the same mutations independently reduce origin initiation efficiency in vivo (*Figure 7*). These observations strongly suggest that reducing the ability to form CtGs recalibrates the response of origins of replication to DDK levels. In the future, it will be of interest to investigate how the levels of DDK and other helicase-activating factors can impact replication origin efficiency in vivo when the Mcm4 and Mcm6 tails are altered.

Our proposed mechanism (*Figure 8*) is analogous to the regulation of Sic1 proteolysis by CDK (*Nash et al., 2001*). Multiple CDK phosphorylation sites on Sic1 are each weakly recognized by the Cdc4 F-box protein, leading to ubiquitinylation and degradation of Sic1. Although weakly bound, the accumulation of multiple phosphorylation sites on Sic1 promotes a switch-like response to CDK levels. We propose that the similar mechanisms are operating during helicase activation, with the Mcm4 and Mcm6 tails providing the multiple phosphorylation-dependent binding sites and the inefficient conversion of the CtG to the CMG performing the role analogous to the suboptimal Cdc4 interaction.

Processes beyond those shown in *Figure 8* are likely to play important roles in origin activation. For example, our experiments did not include phosphatases that could counteract the action of DDK and CDK. Such phosphatases might be recruited by Rif1, a protein that interacts with protein phosphatase 1 and has been implicated in the regulation of replication initiation (*Hafner et al., 2018*; *Hiraga et al., 2014*; *Hiraga et al., 2018*; *Mattarocci et al., 2014*). The presence of phosphatases could regulate Mcm2-7 N-terminal tail phosphorylation levels, further sensitizing origin activation to changes in DDK level. In addition, phosphatases may be essential to remove Mcm2-7-tail phosphorylation after helicase activation to ensure that the tails of active helicases do not continue to compete for limiting helicase-activation factors.

# Materials and methods

## Key resources table

| Reagent type (species) or resource | Designation | Source or reference | Identifiers | Additional information |
|---|---|---|---|---|
| Strain, strain background (*Saccharomyces cerevisiae*) | yCKR006 | This study | *MATa ade2-1 can1-100 pep4Δ*(unmarked) *bar1Δ*(unmarked) *HIS3*::pSKM004 (*GAL1,10-MCM2, Flag-MCM3*), *URA3*::pALS1 (*GAL1,10-CDT1, GAL4*), *LYS2*::pST022 (*GAL1,10-MCM4-SNAP, MCM5*), *TRP1*::pCKR006 (*GAL1,10-MCM6-N-termΔ, MCM7*) | Mcm2-7$^{4SNAP-6ΔN}$ expression strain. Additional information in Materials and methods. Available upon request from Bell Lab. |
| Strain, strain background (*Saccharomyces cerevisiae*) | yCKR009 | This study | *MATa ade2-1 can1-100 pep4Δ*(unmarked) *bar1Δ*(unmarked) *HIS3*::pSKM004 (*GAL1,10-MCM2, Flag-MCM3*), *URA3*::pALS1 (*GAL1,10-CDT1, GAL4*), *LYS2*::pST022 (*GAL1,10-MCM4-SNAP, MCM5*), *TRP1*::pCKR009 (*GAL1,10-MCM6-AD /E+ASP/Q, MCM7*) | Mcm2-7$^{4SNAP-6AD/E+ASP/Q}$ expression strain. Additional information in Materials and methods. Available upon request from Bell Lab. |
| Strain, strain background (*Saccharomyces cerevisiae*) | yCKR047 | This study | *MATa ade2-1 can1-100 pep4Δ*(unmarked) *bar1Δ*(unmarked) *TRP1*::pSKM003 (*GAL1,10-MCM6,MCM7*), *URA3*::pALS1 (*GAL1,10-DT1,GAL4*), *HIS3*::pST048 (*GAL1,10-MCM2-LPETGG, Flag-MCM3*), *LYS2*::pCKR022 (*GAL1,10-MCM4-AD/E+ASP/Q, MCM5*) | Mcm2-7$^{2SORT-4AD/E+ASP/Q}$ expression strain. Additional information in Materials and methods. Available upon request from Bell Lab. |
| Strain, strain background (*Saccharomyces cerevisiae*) | yMM034 | This study | *MATa ade2-1, trp1-1, leu2-3,112, his3-11,15 ura3-1, can1-100, bar1Δ* (unmarked), *LYS2*:: HisG, *pep4Δ* (unmarked), *LEU2*:: (*CDC45*-3xFlag-LPETGG) | Cdc45$^{SORT}$ expression strain. Additional information in Materials and methods. Available upon request from Bell Lab. |

*Continued on next page*

*Continued*

| Reagent type (species) or resource | Designation | Source or reference | Identifiers | Additional information |
|---|---|---|---|---|
| Strain, strain background (*Saccharomyces cerevisiae*) | yST179 | *Ticau et al., 2017* | *MATa ade2-1 can1-100 pep4Δ*(unmarked) *bar1Δ*(unmarked) *TRP1*::pSKM003 (*GAL1,10-MCM6,MCM7*), *URA3*::pALS1 (*GAL1,10-Cdt1,GAL4*), *HIS3*::pST036 (*GAL1,10-SORT-MCM2,Flag-MCM3*), *LYS2*::pSKM002 (*GAL1,10-MCM4, MCM5*) | Mcm2-7²ˢᴼᴿᵀ expression strain. |
| Strain, strain background (*Saccharomyces cerevisiae*) | yST147 | *Ticau et al., 2017* | *MATa ade2-1 can1-100 pep4Δ*(unmarked) *bar1Δ*(unmarked) *TRP1*::pSKM003 (*GAL1,10-MCM6,MCM7*), *HIS3*::pSKM004 (*GAL1,10-MCM2,Flag-MCM3*), *LYS2*::pST022 (*GAL1,10-fSNAP-MCM4, MCM5*), *URA3*::pALS1 (*GAL1,10-CDT1, GAL4*) | Mcm2-7⁴ˢᴺᴬᴾ expression strain. |
| Strain, strain background (*Saccharomyces cerevisiae*) | yML512 | This study, *Randell et al., 2010* | *MatA, ade2-1, ura3-11, his3-11,15, can-100, mcm4Δ*::hisG *TRP1*::pJR140 (*MCM4-HA/HIS-AD/E ASP/Q*) *LEU2*::p405-BrdU-lnc | Additional information in Materials and methods. Available upon request from Bell Lab. |
| Strain, strain background (*Saccharomyces cerevisiae*) | yML513 | This study, *Randell et al., 2010* | *MatA, ade2-1, ura3-11, his3-11,15, can-100, mcm4Δ*::hisG *TRP1*::pJR149 (*MCM6-HA/HIS-AD/E ASP/Q*) *LEU2*::p405-BrdU-lnc | Additional information in Materials and methods. Available upon request from Bell Lab. |
| Strain, strain background (*Saccharomyces cerevisiae*) | yML514 | This study, *Randell et al., 2010* | *MatA, ade2-1, ura3-11, his3-11,15, can-100, mcm4Δ*::hisG *TRP1*::pJR121 (*MCM4-HA/HIS-AD/E ASP/Q*) *LEU2*::p405-BrdU-lnc | Additional information in Materials and methods. Available upon request from Bell Lab. |
| Recombinant DNA reagent | pSNAP-tag (T7)-2 | NEB | N9181S, RRID:Addgene_90314 | Vector that can be used for expression of SNAP-protein alone (20 kDa). |
| Recombinant DNA reagent | pUC19-ARS1 | *Heller et al., 2011* | pUC19 (*ARS1*) | Used in ensemble CMG formation assay and to construct the SM templates. |
| Recombinant DNA reagent | pCKR006 | This study | pSK003 (*GAL1,10-MCM6-N-termΔ, MCM7*) | Additional information in Materials and methods. Available upon request from Bell Lab. |
| Recombinant DNA reagent | pCKR009 | This study | pSKM003 (*GAL1,10-MCM6-AD/E+ASP/Q, MCM7*) | Additional information in Materials and methods. Available upon request from Bell Lab. |
| Recombinant DNA reagent | pCKR022 | This study | pSKM002 (*GAL1,10-MCM4-AD/E+ASP/Q, MCM5*) | Additional information in Materials and methods. Available upon request from Bell Lab. |
| Recombinant DNA reagent | pCKR032 | This study | pMM033 (*CDC45-3xFlag-LPETGG*) | Additional information in Materials and methods. Available upon request from Bell Lab. |

*Continued on next page*

*Continued*

| Reagent type (species) or resource | Designation | Source or reference | Identifiers | Additional information |
|---|---|---|---|---|
| Recombinant DNA reagent | pLDK03 | This study | pFJD5 (6XHIS-PreScission Protease site-Psf3-LPETGG) | Used to express GINS$^{SORT649}$ in bacteria. Additional information in Materials and methods. Available upon request from Bell Lab. |
| Recombinant DNA reagent | pLDK04 | This study | pSNAP-tag (T7)—2 (8xARG-Mcm4 N-terminal tail-SNAP) | Used to express the Mcm4 N-terminal tail fused to the SNAP tag in bacteria. Additional information in Materials and methods. Available upon request from Bell Lab. |
| Recombinant DNA reagent | p405-BrdU-Inc | *Viggiani and Aparicio, 2006* | p405-BrdU-Inc (*LEU2*, BrdU Incorporation), RRID:Addgene_71791 | Plasmid that allows yeast cells to incorporate BrdU. |
| Sequence-based reagent | Oligo for 1.2 kb circular template | IDT | 5′-AATTA**GCGGCCGC**AAGGC/iBiodT/GATTAAGTT-3′ | NotI site in bold. |
| Sequence-based reagent | Oligo for 1.2 kb circular template | IDT | 5′-ATTAA**GCGGCCGC**AGCGGA/iAlexa488N/TAACAATTT-3′ | NotI site in bold. |
| Sequence-based reagent | Oligo for 1.1 kb unwinding template | IDT | 5′-/5Cy5/TACGCCAAGCTTGCATGCGGATGTT**GC**—3′ | Part of NotI sticky end in bold. |
| Sequence-based reagent | Oligo for 1.1 kb unwinding template | IDT | 5′-/5Phos/**GGCCGC**AACATCCGCATGCAAGCTTGGCGTA/3BHQ2/—3′ | Part of NotI sticky end in bold. |
| Peptide, recombinant protein | Peptide for coupling to dyes for Sortase labeling | *Ticau et al., 2017* | NH$_2$-GGGHHHHHHHHHHHC-COOH | |
| Chemical compound, drug | DY549-P1 | Dyomics | Dyomics: 549P1-03 | Maleimide-coupled fluorescent dye |
| Chemical compound, drug | DY649-P1 | Dyomics | Dyomics: 649P1-03 | Maleimide-coupled fluorescent dye |
| Chemical compound, drug | SNAP-Surface 549 | NEB | | Fluorescent substrate used to label SNAP-tag fusion proteins (Mcm2-) |
| Antibody | Anti-Mcm2-7, rabbit polyclonal | Bell Lab | UM174 | (1:10,000) |
| Antibody | Anti-Cdc45, rabbit polyclonal | *Lõoke et al., 2017* | HM7135 | (1:2000) |
| Antibody | Anti-GINS, rabbit polyclonal | *Lõoke et al., 2017* | HM7128 | (1:2000) |
| Gene (*Saccharomyces cerevisiae*) | MCM2 | *Saccharomyces* Genome Database | SGD: S000000119 | |
| Gene (*Saccharomyces cerevisiae*) | MCM3 | *Saccharomyces* Genome Database | SGD: S000000758 | |
| Gene (*Saccharomyces cerevisiae*) | MCM4 | *Saccharomyces* Genome Database | SGD: S000006223 | |
| Gene (*Saccharomyces cerevisiae*) | MCM5 | *Saccharomyces* Genome Database | SGD: S000004264 | |
| Gene (*Saccharomyces cerevisiae*) | MCM6 | *Saccharomyces* Genome Database | SGD: S000003169 | |
| Gene (*Saccharomyces cerevisiae*) | MCM7 | *Saccharomyces* Genome Database | SGD: S000000406 | |

*Continued on next page*

*Continued*

| Reagent type (species) or resource | Designation | Source or reference | Identifiers | Additional information |
|---|---|---|---|---|
| Gene (*Saccharomyces cerevisiae*) | *CDC45* | *Saccharomyces* Genome Database | SGD: S000004093 | |
| Gene (*Saccharomyces cerevisiae*) | *SLD5* | *Saccharomyces* Genome Database | SGD: S000002897 | |
| Gene (*Saccharomyces cerevisiae*) | *PSF1* | *Saccharomyces* Genome Database | SGD: S000002420 | |
| Gene (*Saccharomyces cerevisiae*) | *PSF2* | *Saccharomyces* Genome Database | SGD: S000003608 | |
| Gene (*Saccharomyces cerevisiae*) | *PSF3* | *Saccharomyces* Genome Database | SGD: S000005506 | |
| Software, algorithm | Matlab | Mathworks | | The 'intervals' files are readable by the imscroll program: (https://github.com/gelles-brandeis/CoSMoS_Analysis; *Gelles and Friedman, 2021*; copy archived at swh:1:rev:3eec2cbfa54018389fc1905b54c4b062723a5a7f) |

## Preparation of unlabeled proteins

Wild-type unlabeled Mcm2-7/Cdt1 and ORC complexes were purified as described previously *Kang et al., 2014*. Wild-type Cdc6 was purified as described in *Frigola et al., 2013*. Wild-type DDK, S-CDK, Sld3/7, Sld2, unlabeled Cdc45, Dpb11, unlabeled GINS, Mcm10, Pol ε, and RPA were purified as described in *Lõoke et al., 2017*.

## Preparation of GINS^SORT649^

Plasmid pFJD5 (gift from K. Labib) was modified with a His-Tag followed by a PreScission Protease tag at the N-terminus of Psf3 and a LPTEGG at the C-terminus of Psf3 (pLDK03, Key Resources Table) to facilitate peptide addition by Sortase. The plasmid was expressed in BL21 (D3) competent *Escherichia coli* (NEB) and purified as described in *Yeeles et al., 2015* with the following modifications. After eluting from HisPur Ni-NTA resin (ThermoFisher Scientific), the eluates were incubated with excess PreScission Protease at 4°C overnight, then incubated with HisPur Ni-NTA resin to remove any uncleaved GINS. The unbound protein was collected and applied to MonoQ column as described in *Yeeles et al., 2015*. The peak GINS fractions were collected and Sortase was used to attach the peptide NH2-GGGHHHHHHHHHHC-COOH coupled to maleimide-Dy649P1 (Dyomics) to the C-terminus of Psf3 to form GINS^SORT649^. Sortase coupling of the fluorescent peptide was performed as described in *Ticau et al., 2017*. After fluorescent labeling, GINS^SORT649^ was applied to a Superdex S200 column as described in *Yeeles et al., 2015*. Peak fractions were pooled and applied to Ni-NTA resin pre-equilibrated with buffer H (25 mM HEPES-KOH [pH 7.6], 5 mM MgOAc, 10% glycerol, and 0.02% NP-40), supplemented with 200 mM potassium acetate (KOAc), 0.02% Nonidet P-40 (NP-40), and 10 mM imidazole, for 30 min with rotation at 4°C to separate peptide-coupled GINS from uncoupled GINS. The resin was washed with buffer H, supplemented with 200 mM KOAc, 0.02% NP-40, and 30 mM imidazole. Fluorescently labeled GINS^SORT649^ was eluted using buffer H, supplemented with 200 mM KOAc, 0.02% NP-40, and 250 mM imidazole. Peak fractions were pooled, aliquoted, and stored at −80°C.

## Preparation of Cdc45^SORT649^

A *CDC45*-expressing plasmid was modified by appending sequences encoding a Flag-epitope followed by LPETGG at the 5′ end of the gene of plasmid pMM032 (Key Resources Table). This plasmid was integrated into genome of yMH109 to create yMM034 (Key Resources Table). This strain was used to purify Cdc45 as described in *Lõoke et al., 2017* with the following modifications. After elution from anti-Flag M2 affinity gel (Sigma), the eluate was attached with Sortase to the peptide NH2-GGGHHHHHHHHHHC-COOH coupled to maleimide-Dy649P1 (Dyomics) as described in

*Ticau et al., 2017*. The modified protein was applied to a Superdex S75 column equilibrated with buffer H, supplemented with 0.3 M KGlut and 5% glycerol. The peak fractions were applied to a His-Pur Ni-NTA resin (ThermoFisher Scientific) equilibrated with buffer H, supplemented with 300 mM KGlut and 5 mM imidazole, for 30 min with rotation at 4°C to separate peptide-coupled from uncoupled Cdc45. The flow-through was discarded and the resin was washed with buffer H, supplemented with 300 mM KGlut and 20 mM imidazole. Fluorescently labeled Cdc45[SORT649] was eluted using buffer H, supplemented with 300 mM KGlut and 250 mM imidazole. Peak fractions were pooled, aliquoted, and stored at −80°C.

## Preparation of Mcm2-7[4SNAP549], Mcm2-7[2SORT549] Mcm2-7[4SNAP549ΔN], Mcm2-7[4SNAP549-6AD/E+ASP/Q], and Mcm2-7[2SORT549-4AD/E+ASP/Q]

The Mcm6-Mcm7 overexpression plasmid was modified by deleting the sequences encoding Mcm6 amino acids 2–105 to create the plasmid pCKR006 (Key Resources Table). This plasmid was integrated into the genome of a strain that contained expression constructs for Mcm2, Mcm3, Mcm4, Mcm5, and Cdt1 to create the yeast strain yCKR006 (Key Resources Table). The Mcm6-Mcm7 overexpression plasmid was modified by substituting serine and threonine in SS/T-P/Q and S/T-D/E sequences to alanine to create the plasmid pCKR009 (Key Resources Table). This plasmid was integrated into the genome of a strain that contained expression constructs for Mcm2, Mcm3, Mcm4, Mcm5, and Cdt1 to create the yeast strain yCKR009 (Key Resources Table). The Mcm4-Mcm5 overexpression plasmid was modified by substituting serine and threonine in SS/T-P/Q and S/T-D/E sequences to alanine to create the plasmid pCKR022 (Key Resources Table). This plasmid was integrated into the genome of a strain that contained expression constructs for Mcm2, Mcm3, Mcm6, Mcm7, and Cdt1 to create the yeast strain yCKR047 (Key Resources Table). Mcm2-7/Cdt1 SNAP-tag ΔN6 (yCKR006), Mcm2-7/Cdt1 SNAP-tag 6AD/E+ASP/Q (yCKR009), Mcm2-7/Cdt1 SORT-tag 4AD/E +ASP/Q (LPETGG at the C-terminus of Mcm2 for fluorescent labeling, yCKR047), Mcm2-7/Cdt1 SORT-tag (LPETGG at the C-terminus of Mcm2 for fluorescent labeling, yST179, Key Resources Table), and Mcm2-7/Cdt1 SNAP-tag (SNAP-tagged at Mcm4 for fluorescent labeling, yST147, Key Resources Table) were purified from the indicated yeast strains as described previously in *Kang et al., 2014* with the following modifications. After ultracentrifugation, the whole cell extract was applied to 1 mL of anti-Flag M2 affinity gel (Sigma) pre-equilibrated with buffer H, 200 mM KGlut, 0.01% NP-40, and 1 mM ATP and incubated with rotation for 2 hr at 4°C. The flow-through was discarded and the resin was washed with 30 mL of buffer H, supplemented with 300 mM KGlut, 0.01% NP-40, and 1 mM ATP. Mcm2-7/Cdt1 was eluted with buffer H, supplemented with 300 mM KGlut, 0.01% NP-40, 1 mM ATP, and 0.1 mg/mL 3xFLAG peptide for the first elution and with 0.3 mg/mL 3xFLAG peptide for the consecutive elutions. After elution from anti-Flag M2 affinity gel, the eluate was fluorescently labeled as follows. For Mcm2-7[4SNAP549], Mcm2-7[4SNAP549-6AD/E + ASP/Q], and Mcm2-7[4SNAP549ΔN] (SNAP at the N-terminus of Mcm4), the FLAG eluate was labeled with SNAP-Surface549 (NEB) by incubating 3x molar excess of dye at 4°C overnight. After coupling the protein to the fluorophore, the reaction was applied to a Superdex S200 column equilibrated with buffer H, supplemented with 300 mM KGlut, 0.01% NP-40, and 1 mM ATP. Peak fractions containing Mcm2-7[4SNAP549], Mcm2-7[4SNAP549ΔN], or Mcm2-7[4SNAP549-6AD/E+ASP/Q]/Cdt1 were pooled, aliquoted, and stored at −80°C. Mcm2-7[4SNAP549]/Cdt1 was used in all experiments with Cdc45[SORT649], with exception of experiments with N-terminal mutants of Mcm2-7.

For Mcm2-7[2SORT549 or 2SORT549-4AD/E+ASP/Q]/Cdt1 (LPETGG at the C-terminus of Mcm2), the FLAG eluate was attached with Sortase to the peptide NH2-GGGHHHHHHHHHHC-COOH coupled to maleimide-Dy649P1 (Dyomics) as described in *Ticau et al., 2017*. Mcm2-7[2SORT549]/Cdt1 was applied to a Superdex S200 column equilibrated with buffer H, supplemented with 300 mM KGlut, 0.01% NP-40, and 1 mM ATP. Peak fractions containing fluorescently labeled Mcm2-7[2SORT549 or 2SORT549-4AD/E + ASP/Q]/Cdt1 were pooled, aliquoted, and stored at −80°C. Mcm2-7[2SORT549]/Cdt1 was used in all experiments with GINS[SORT649].

## Determining Mcm2-7, Cdc45, and GINS labeling fraction

To determine what fraction of Mcm2-7[4SNAP549] were fluorescently labeled, SNAP-Surface549 labeled Mcm2-7 was mixed with maleimide-DY-649P1 dissolved in anhydrous dimethyl sulfoxide (DMSO) in a 1:1 ratio, and the reaction and analysis were carried out as described in *Ticau et al., 2015*. The

labeled fraction for Mcm2-7[4SNAP549] was determined to be 0.74 ± 0.05. To determine the labeling fraction of Cdc45[SORT649] (and GINS[SORT649]), the labeled proteinwas mixed with maleimide-DY549P1 dissolved in anhydrous DMSO in a 1:1 molar ratio at 4°C for 10 min. The maleimide-DY549P1 will label free cysteines in the protein. The reaction was terminated with 2 mM DTT. We added 5 nM of maleimide-DY549P1-labeled Cdc45[SORT649] (or 8 nM maleimide-DY549P1-labeled GINS[SORT649]) to the SM CMG-formation assay with 1.3 nM DDK to obtain single Cdc45[SORT649] (or GINS[SORT649]) traces. The fraction of maleimide-DY549P1-labeled Cdc45[SORT649] (or GINS[SORT649]) that also contained DY-649P1 was determined and reported as the percent labeling by the DY-649P1 (we assume that coupling of maleimide-DY-549P1 or DY-649P1 to Cdc45, Mcm2-7, and GINS is not influenced by the presence or absence of the 649 or 549 label). The labeled fractions for Cdc45[SORT649] and GINS[SORT649] were determined to be 0.73 ± 0.07 and 0.71 ± 0.07, respectively. To lower the labeling fraction of Cdc45[SORT649] to 0.036, the Cdc45[SORT649] preparation was mixed at a 1:19 ratio with unlabeled Cdc45.

## Generation of circular template

To create biotin- and fluorescently labeled circular origin DNA, a 1.3 kb ARS1-containing interval of the DNA plasmid template pUC19-ARS1 was amplified with NotI-site-containing primers, one of which contained a biotin and the other an Alexa-Fluor-488 dye (Key Resources Table). After PCR cleanup with QIAquick PCR Purification Kit (Qiagen), the PCR product was digested with NotI (NEB) at 37°C for 4 hr and repurified with a QIAquick PCR Purification Kit. We next performed a ligation with 0.2 ng/µL of digested DNA and 0.04 U/µL of T4 DNA ligase (NEB) at 18°C overnight to favor intramolecular ligation. The ligation products were purified by phenol–chloroform extraction and concentrated by ethanol precipitation. The concentrated DNA was then run on a 1.5% TBE-agarose gel, and the circular DNA band was extracted and purified with QIAquick Gel Extraction Kit (Qiagen).

## SM assay for CMG formation

Biotinylated Alexa-Fluor-488-labeled, 1.2-kb-long circular-DNA molecules containing an origin were coupled to the surface of a reaction chamber through streptavidin. We identified DNA molecule locations by acquiring 4–7 images with 488 nm excitation at the beginning of the experiment. Helicase loading reaction was performed as described in *Ticau et al., 2015*. Reaction buffers for helicase loading were as described in *Ticau et al., 2015*. All subsequent steps (i.e., Mcm2-7 phosphorylation and CMG formation) used the same helicase-loading buffer. After helicase loading, either 1.3 or 6.5 nM DDK was added. Chambers were then washed with two chamber volumes of buffer A (25 mM HEPES-KOH [pH 7.6], 5 mM magnesium acetate [MgOAc], 0.02% NP-40), supplemented with 0.5 M NaCl (HSW1), followed by one chamber volume of buffer A supplemented with 300 mM KGlut. Next, CMG-formation reaction was added containing 7.5 nM CDK, 15.5 nM Sld2, 10 nM Dpb11, 7.5 nM Pol ε, 50 nM GINS, 12.5 nM Sld3/7, and 12.5 nM Cdc45. DNA was imaged immediately after adding the CMG-formation reaction to the slide but not throughout the experiment. When using Cdc45[SORT649] or GINS[SORT649], they were added at 5 nM and 7 nM, respectively. After ~30 min, chambers were washed with four chamber volumes of buffer A supplemented with 0.5 M NaCl (HSW2), followed by one chamber volume of buffer A supplemented with 300 mM KGlut.

## SM assay for DNA unwinding

To obtain the 1.1 kb linear-DNA molecules used for the DNA unwinding assay, the same origin-containing interval of pUC19-ARS1 was amplified with a NotI-site-containing primer and a biotin-containing primer (Key Resources Table). The PCR-amplified DNA was digested with NotI (NEB), purified with a QIAquick PCR Purification Kit (Qiagen), and ligated with oligos each containing a Cy5- or BHQ-2 label and NotI sticky ends complimentary to the amplified DNA template. After ligation, the DNA product was purified with a QIAquick PCR Purification Kit (Qiagen). The DNA was coupled to the surface of a reaction chamber in a streptavidin-mediated reaction to slide-attached PEG-biotin as described previously in *Ticau et al., 2015*.

Helicase-loading, DDK, CMG-formation, and DNA-unwinding reactions were performed as described above with the following modifications: for DNA unwinding, 12.5 nM RPA and 0.625 nM Mcm10 were added. No high-salt wash was performed after DNA-unwinding reaction.

## SM microscopy

The micro-mirror total internal reflection microscope used for multiwavelength SM imaging using excitation wavelengths 488, 532, and 633 nm is described in *Friedman et al., 2006*. Briefly, the chamber surface was cleaned and derivatized using a 200:1 ratio of silane-NHS-PEG and silane-NHS-PEG-biotin as described in *Ticau et al., 2015*. Biotinylated Alexa-Fluor-488-labeled 1.2 kb circular DNA molecules containing an origin were coupled to the surface of a reaction chamber through streptavidin. Subsequently, Mcm2-7 loading, Mcm2-7 phosphorylation, and CMG-formation reactions (or DNA unwinding) were added to a chamber of the microscope slide. DNA was imaged immediately after adding the CMG-formation reaction to the slide but not throughout the experiment. After addition of CMG-formation proteins to Mcm2-7-bound DNA, images were acquired according to one of two different protocols. For *Figure 3* through 6, we alternated acquisition of 1 s duration frames with 532 nm excitation (0.4 mW power incident to the final focusing lens) with 1 s duration frames with 633 nm excitation (0.25 mW incident power). For *Figure 2*, frames of 1 s duration were acquired continuously with both 532 nm (0.4 mW incident power) and 633 nm (0.30 mW incident power) excitation. In both protocols, a computer-controlled focus adjustment utilizing a 785 nm laser was performed approximately once every second (*Crawford et al., 2008*). For all experiments, we used the following filters: Semrock NF03-532E-25 (notch filter at 532 nm to remove scatter during 532 nm laser excitation), Semrock NF03-633E-25 (notch filter at 633 nm to remove scatter during 633 nm laser excitation), and Semrock FF01-525/50 (bandpass filter for removing scatter during 488 nm laser excitation). This cycle was repeated over the course of an ~30 min experiment. After ~30 min of incubation, chambers were washed with four chamber volumes of buffer H supplemented with 500 mM NaCl and one chamber volume of buffer H supplemented with 300 mM of KGlut. Recording of protein photobleaching was then performed as described in *Ticau et al., 2015*.

## SM data analysis

Analysis of the CoSMoS data sets was as described in *Ticau et al., 2015*. Records of protein SM fluorescence were corrected for background fluorescence as follows: one set of intensity time records summed intensities of 5 × 5 pixel areas centered on the DNA molecule locations ($D_j(m)$ records for integrated intensity from the $m$th frame for the $j$th DNA molecule area). Additional time records similarly recorded time-smoothed background intensity levels from areas adjacent to each DNA molecule location ($B_j(m)$ background records for the $j$th DNA molecule) and the time-smoothed background offset record for the camera with no light incident ($C(m)$ record). Background-corrected fluorescence-intensity records were calculated as $D_j(m) - B_j(m)$.

Background fluorescence (solution dye + slide autofluorescence) results in a ($B_j(m) - C(m)$) value that is proportional to the local laser excitation at the $j$th DNA site. One DNA site record was chosen as having a reference level of laser excitation ($B_R(m) - C(m)$), and the time record for all other DNA sites was background-corrected and normalized to correspond to the laser excitation at that reference $j$ = R site. Accordingly, this background-corrected normalized time record $N_j(m)$ for each DNA site is

$$N_j(m) = \left(D_j(m) - B_j(m)\right) \frac{\left(B_R(m) - C(m)\right)}{\left(B_j(m) - C(m)\right)}$$

Source data for the SM experiments is provided as intervals files in Zenodo under the accession code 4309997. The 'intervals' files can be read and manipulated by the Matlab program imscroll, which is publicly available: https://github.com/gelles-brandeis/CoSMoS_Analysis.

## Intensity histogram fitting

Using a noise (variance) vs. (signal mean) measurement, we determined the calibration factor $k$, the number of photons corresponding to one digitized intensity unit (analog-to-digital unit [ADU]) ($k$ = 0.0118 photons/ADU) (*Friedman et al., 2006*). From the recording of each experiment, we defined $g = \langle(B_R(m) - C(m))\rangle k$, the mean value (in photons) of the reference DNA site background intensity averaged over all frames for that experiment.

The intensity data $N_j(m)$ from $j$ = 200 to 400 DNA locations over $m$ = 600 time points (1500 s recording duration) were pooled for intensity distribution analysis of $N$. Outlier measurements with

$N$ values > 50,000 ADU (in all experiments less than 5% of measurements) were excluded from histograms and fits.

The distributions of corrected intensities $N$ from each experiment were fit to a modified binomial-distributed Gaussian mixture model probability density function

$$p(N) = \sum_{i=0}^{n} \left\{ \frac{P_i^{'}}{\sigma_i \sqrt{2\pi}} \exp\left( -\frac{1}{2} \left[ \frac{N - is}{\sigma_i} \right]^2 \right) \right\}$$

with the normalized amplitudes

$$P_i^{'} = P_i / \sum_{i=0}^{n} P_i$$

derived from the modified Binomial distribution

$$P_i = \begin{cases} q + \binom{n}{i} \lambda^i (1-\lambda)^{n-i} & i = 0 \\ \binom{n}{i} \lambda^i (1-\lambda)^{n-i} & i > 0 \end{cases}$$

and the component widths

$$\sigma_i = \sqrt{ \left( \frac{\sqrt{g + isk}}{k} \right)^2 + (xi)^2 }$$

with fit parameters $q$, $s$, $\lambda$, and $x$. The number of components was held fixed at $n = 8$, although component numbers for $i > 6$ did not contribute significantly to the probability density function ($P_i^{'} <$ 0.01) in any of the data sets analyzed (*Supplementary file 1a* and *Supplementary file 1b*). In this model, $q$ is an additional amplitude in the zeroth peak to account for target molecules incapable of binding labeled protein, $s$ is the corrected intensity (expressed in ADU) corresponding to a single dye molecule, $n\lambda$ is the mean number of dye molecules per binding-competent Mcm2-7-bound DNA, and $x$ is an additional variation in fluorescent spot intensity (expressed in ADU) presumed to arise from DNA Brownian motion. In some experiments with minimal labeled protein binding (marked in *Supplementary file 1a, b*), the data were not sufficient to constrain the confidence interval lower limit for one or both variables $\lambda$ and $x$. Fit parameters were determined by maximizing the likelihood using customized Matlab software (in https://github.com/gelles-brandeis/jganalyze; *Friedman and Gelles, 2015*).

## Bulk CMG-formation assay
Ensemble CMG-formation assays were performed as described in *Champasa et al., 2019*.

## Phosphorylation assays on SuperSep Phos-tag gels
Each incubation step was performed in a thermomixer (Eppendorf) with shaking at 1250 rpm at 25° C. The DNA template was the same used in CMG-formation assays as described in *Champasa et al., 2019*. Helicase loading reactions contained 100 nM Mcm2-7, 45 nM ORC, and 45 nM Cdc6. After loading, DDK was added at varying concentrations (0 nM, 20 nM, 80 nM, 120 nM, 260 nM, and 500 nM) for 20 min. The supernatant was removed by applying the reaction to a DynaMag-2 magnet (ThermoFisher Scientific). Reactions were washed with buffer H, 300 mM KCl, and 0.01% NP-40 three times. Proteins were released from the DNA by incubation with 5 U of DNA I (Worthington) in 10 mL of 25 mM HEPES-KOH (pH 7.6), 5 mM MgOAc, 200 mM NaCl, 5% glycerol, 0.02% NP-40, and 2 mM $CaCl_2$ for 20 min at 25°C before running on a pre-cast 10% SuperSep Phos-tag gel at 25 mA for > 2 hr. The gels were then stained using Krypton Protein Stain (ThermoFisher Scientific).

When Mcm4 N-terminal tail fused to SNAP-tag protein was used, then 500 nM of protein was phosphorylated with varying concentrations of DDK (0 nM, 20 nM, 80 nM, 120 nM, 260 nM, and 500 nM) for 20 min. Reactions were run on a pre-cast 10% SuperSep Phos-tag gel at 25 mA for >2 hr. The gels were then stained using Krypton Protein Stain (ThermoFisher Scientific).

## BrdU ChIP-seq

Fifty milliliters of OD 0.5 cell culture in YPD was used for each sample. Initially, cells were arrested in G1 phase by supplementing the media with 5 nM α-factor for 2.5 hr. The α-factor-arrested cells were washed once with MilliQ water and resuspended in YPD release media supplemented with 50 mM HU, 1.3 mM BrdU (Millipore Sigma), and 300 μg/mL of pronase (Millipore Sigma). After 1 hr, cell growth and replication were terminated by adding sodium azide (0.1% final concentration) to the samples. For DNA extraction, cells were washed once with MilliQ water (supplemented with 0.1% sodium azide), followed by degradation of cell walls by zymolyase 20T (Sunrise Science Products) (100 μg/mL final concentration) in 10 mM Tris-HCl pH 8, 10 mM CaCl₂, 1 M Sorbitol, 0.1% 2-mecaptoethanol, and 0.1% sodium azide 1 hr at 30°C. Following cell wall degradation, spheroplasts were collected by centrifugation at 200 × $g$ for 5 min and lysed in 50 mM Tris-HCl pH 8, 10 mM EDTA, and 1.2% SDS. Cell lysates were brought to 150 mM NaCl and 100 μg/mL RNase A (Amresco) and incubated for 2 hr at 37°C. Following RNA degradation, Proteinase K (Amresco) was added to the lysate at 50 μg/mL final concentration and incubated for an additional hour at 65°C. Finally, samples were extracted with phenol/chloroform and residual phenol was removed by an additional chloroform extraction. To precipitate the resulting purified DNA, sodium acetate was added to a final concentration of 300 mM, followed by the addition of three volumes of 100% ethanol. After centrifugation at 15,000 × $g$ for 15 min at room temperature, the resulting pellet was resuspended in 400 μL of water. The purified DNA was fragmented to ~250 bp average length by sonication and heat denatured (5 min at 95°C), followed by snap-cooling in ice water. For BrdU immunoprecipitation (IP), 400 μL of IP buffer (50 mM HEPES, pH 7.5, 140 mM NaCl, 1 mM EDTA, 1% Triton X-100, 0.1% sodium deoxycholate) and 3 μL of anti-BrdU antibody (BD; catalog: 555627) were added to the DNA and incubated for 2 hr at room temperature. Antibody-DNA complexes were collected by the addition of 20 μL of Sepharose G beads and incubated for 1 hr at room temperature. DNA-bound beads were washed three times with IP buffer, followed by two sequential rounds of elution with elution buffer (100 mM Tris-HCl pH 7.5, 1 mM EDTA, 1% SDS, 400 mM NaCl) at 65°C for 15 min. The eluted DNA was ethanol precipitated as described, and the DNA pellet was resuspended in 100 μL of water. Sequencing libraries were prepared using NEBNext Ultra II DNA Library Prep Kit for Illumina, according to the manufacturer's instructions, and sequenced using the Illumina HiSeq2000 platform. Sequencing data associated with the in vivo replication data has been deposited in Zenodo under the accession code 4507881.

## Analysis of genome sequencing data

Raw sequencing reads were mapped to the *Saccharomyces cerevisiae* genome (S288C reference) using Bowtie2, sorted and indexed with Samtools. Only uniquely mapping reads were used for analysis. To normalize the data, count per kilobase million (cpm) value was calculated for each base as (read count for base/all reads) × $10^3$ (kilobase) × $10^6$ (million). This value was next scaled by dividing it with the lower 10% quantile (non-replicating areas of the genome) for each barcode to obtain the scaled cpm (s-cpm) value. The s-cpm value was smoothed using one-dimensional Gaussian filter with a sigma value of 1000 and plotted for every bp on chromosome X. To plot the averaged s-cpm values for all annotated replication origins, replication origin positions were downloaded from https://yeastmine.yeastgenome.org/ using InterMine (*Smith et al., 2012*). Next, replication origins were aligned based on their midpoints, and the average of all s-cpm values was calculated from −5000 bp to +5000 bp from origin midpoints. Early origins were defined as origins with a time of replication less than 20 min (*Siow et al., 2012*; *Yabuki et al., 2002*).

## Acknowledgements

We are grateful to Alexandra Pike and Shalini Gupta for comments on the manuscript, Karim Labib for providing the plasmid pFJD5, and Michael Maloney for making the Cdc45 labeling construct.

## Additional information

### Funding

| Funder | Grant reference number | Author |
| --- | --- | --- |
| Howard Hughes Medical Institute | Investigator | Stephen P Bell |
| National Institutes of Health | R01 GM52339 | Stephen P Bell |
| National Institutes of Health | R01 GM81648 | Jeff Gelles |
| National Institutes of Health | T32 GM007287 | Lorraine De Jesus Kim |

The funders had no role in study design, data collection and interpretation, or the decision to submit the work for publication.

### Author contributions

Lorraine De Jesús-Kim, Conceptualization, Data curation, Formal analysis, Validation, Investigation, Visualization, Methodology, Writing - original draft, Writing - review and editing; Larry J Friedman, Conceptualization, Data curation, Software, Formal analysis, Supervision, Validation, Methodology, Writing - review and editing; Marko Lõoke, Data curation, Formal analysis, Investigation, Methodology, Writing - review and editing, Performed in vivo DNA replication analysis; Christian K Ramsoomair, Resources, Validation; Jeff Gelles, Conceptualization, Software, Formal analysis, Supervision, Funding acquisition, Validation, Investigation, Visualization, Methodology, Project administration, Writing - review and editing; Stephen P Bell, Conceptualization, Supervision, Funding acquisition, Investigation, Visualization, Methodology, Writing - original draft, Project administration, Writing - review and editing

### Author ORCIDs

Larry J Friedman  http://orcid.org/0000-0003-4946-8731
Jeff Gelles  https://orcid.org/0000-0001-7910-3421
Stephen P Bell  https://orcid.org/0000-0002-2876-610X

### Decision letter and Author response

Decision letter https://doi.org/10.7554/eLife.65471.sa1
Author response https://doi.org/10.7554/eLife.65471.sa2

## Additional files

### Supplementary files

• Supplementary file 1. Flourescence-intensity histogram fit parameters. Supplementary file 1a. Fit parameters for Cdc45 fluorescence-intensity histograms. Supplementary file 1b. Fit parameters for GINS fluorescence-intensity histograms.

• Transparent reporting form

### Data availability

Source data for the single-molecule experiments is provided as "intervals" files that can be read and manipulated by the Matlab program imscroll, which is publicly available: https://github.com/gelles-brandeis/CoSMoS_Analysis (copy archived at https://archive.softwareheritage.org/swh:1:rev:3eec2cbfa54018389fc1905b54c4b062723a5a7f/). The source data are archived as doi: https://doi.org/10.5281/zenodo.4309997. The sequencing data from the in vivo replication assay is archived as DOI: https://doi.org/10.5281/zenodo.4507881.

The following datasets were generated:

| Author(s) | Year | Dataset title | Dataset URL | Database and Identifier |
|---|---|---|---|---|
| Looke M, Bell SP | 2021 | Origins of replication in vivo source data | https://doi.org/10.5281/zenodo.4507881 | Zenodo, 10.5281/zenodo.4507881 |
| Jesus-Kim LD, Friedman L, Bell SP, Gelles J | 2021 | Single-molecule source data files | https://doi.org/10.5281/zenodo.4309997 | Zenodo, 10.5281/zenodo.4309997 |

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
