## [Decision Letter]

**Acceptance summary:**

The manuscript provides new and convincing evidence that a heretofore unknown intermediate state for replication start contains multiple copies of the GINS and Cdc45 proteins prior to initiation at each origin with one double hexamer of the MCM2-7 complex. The number of GINS and Cdc45 is determined by DDK phosphorylation of the MCM's and the probability to create an active helicase (CMG) is increased with multiple numbers of the bound ancillary factors. Furthermore the key issues raised by the reviewers have been adequately dealt with in this revision. The single molecule studies and biochemistry are beautifully executed providing the evidence for such intermediates. We agree that the addition of in vivo studies demonstrates that modulating the multiplicity of DDK phosphorylation (and proposed, CtG formation) has an impact on origin usage in cells.

**Decision letter after peer review:**

Thank you for submitting your article "DDK regulates replication initiation by controlling the multiplicity of Cdc45-GINS binding to Mcm2-7" for consideration by *eLife*. Your article has been reviewed by three peer reviewers, including Michael R Botchan as the Reviewing Editor and Reviewer #1, and the evaluation has been overseen by Kevin Struhl as the Senior Editor. The following individual involved in review of your submission has agreed to reveal their identity: Dirk Remus (Reviewer #3).

The reviewers have discussed the reviews with one another and the Reviewing Editor has drafted this decision to help you prepare a revised submission.

We would like to draw your attention to changes in our policy on revisions we have made in response to COVID-19 (https://elifesciences.org/articles/57162). Specifically, when editors judge that a submitted work as a whole belongs in *eLife* but that some conclusions require a modest amount of additional new data, as they do with your paper, we are asking that the manuscript be revised to either limit claims to those supported by data in hand, or to explicitly state that the relevant conclusions require additional supporting data.

All of the reviewers concurred that the manuscript provides important new data and hope that the comments below will help in improving the paper before publication.

Summary:

De Jesus-Kim and colleagues use co-localization single-molecule spectroscopy to study the formation of the CMG replicative holo-helicase at replication origins using purified proteins. They identify a new intermediate (CtG, Cdc45-tail-GINS). This intermediate is en route to CMG formation and includes Cdc45 and GINS, which are recruited by the N-terminal tails of Mcm4 and Mcm6, in a process that depends on phosphorylation by DDK. It is established that multiple phospho-sites exist both in N-Mcm4 and N-Mcm6. The authors find that the degree of phosphorylation controls the number of Cdc45 and GINS molecules that can be recruited onto a loaded MCM at any one time. Furthermore, the authors find that the transition from CtG to CMG is inefficient, however, CtGs that contain a higher number of Cdc45 and GINS particles convert to CMG more frequently. This is an elegant, technically sound study that informs the mechanism of eukaryotic origin activation and will be of interest to a broad readership.

The three reviewers all have suggestions for your consideration in the revisions. However, we feel that the following are the issues that must be addressed either by further text revisions or experiments:

1) The Abstract is difficult to understand and perhaps misleading, especially for those outside the field. The actual protein composition of the CtGs is not known from the data and the authors are of course clear about this in the manuscript. The Abstract should clarify this.

2) Understood that the deletion of the Mcm 4 "tail" is lethal but is that also true in their system? Would that deletion with a wt Mcm6 lead to a more dramatic loss in the measures used?

3) Figure 1B. Can the authors explain why the number of molecules observed in DDK/Sld3/lambda PP is >13 fold lower than n in DDK/Sld3?

4) Figure 2. In panel A the intensities fluctuation is high, suggesting that up to four Cdc45 molecules bind at a time. Conversely, when SDPGC is present variation is greatly reduced and only up to 1 or 2 molecules of Cdc45 bind. Do SDPGC prevent non-specific interactions? This could be discussed.

5) Do the authors only analyse loaded MCM molecules that are double-hexameric? See the representative photobleaching curve in Figure 2A. Can the authors explain whether or not only particles that contain 2 fluorescently labelled MCM complexes are included in the analysis?

6) Have the authors attempted to omit Mcm10 from the CMG formation assay? And could they observe any change in dye emission?

7) In the subsection “Only a subset of the multiple Mcm2-7-bound Cdc45 proteins form CMGs”, the authors conclude that the two CMGs can assemble independently at each MCM DH. The statistical data for this conclusion are not convincing. Can the authors exclude that the 41% observed vs. 57% expected MCM DHs with two salt-resistant labeled Cdc45 molecules is not due to an error in the estimation of the Cdc45 labeling efficiency? Also, in the case of 1.3 nM DDK only 7 molecules were analyzed, which seems too low a number to support any statistically significant conclusion. As the question of coordination between CMG assembly at the MCM DH is mechanistically very important, the authors may want to consider to tone down the conclusion that CMGs can assembly independently or provide additional data in support of their conclusion.

8) The authors demonstrate that GINS binding to MCMs is dependent on Sld3/7 and Cdc45. Is GINS binding dependent on Sld2, Dpb11, Pol-epsilon, and CDK as well? Does elimination of any one of these factors differentially affect GINS recruitment and CMG formation? This could point to the mechanism of CtG to CMG transformation. The authors are well-positioned to address this question and may have done so already.

9) Figure 4—figure supplement 2B: How was this experiment performed. The text states that protein was stained with Krypton protein stain. Where are the other MCM subunits then? Or was this experiment performed with isolated Mcm4 tail? Please clarify.

---

## [Author Response]

[…] The three reviewers all have suggestions for your consideration in the revisions. However, we feel that the following are the issues that must be addressed either by further text revisions or experiments:1) The Abstract is difficult to understand and perhaps misleading, especially for those outside the field. The actual protein composition of the CtGs is not known from the data and the authors are of course clear about this in the manuscript. The Abstract should clarify this.

We have modified the Abstract to include the following clarification: “Initially, Cdc45, GINS, and likely additional proteins are recruited to unstructured Mcm2-7 N-terminal tails in a Dbf4-dependent kinase (DDK)-dependent manner, forming Cdc45-tail-GINS intermediates (CtGs).”

2) Understood that the deletion of the Mcm 4 "tail" is lethal but is that also true in their system? Would that deletion with a wt Mcm6 lead to a more dramatic loss in the measures used?

We have not tested an Mcm4-tail deletion although the reviewer is correct, it could function in the biochemical context of our assays. Instead, we have provided a new set of experiments that we believe are a better test of our model that reducing the multiplicity of CtG formation reduces CMG formation. We have produced labeled Mcm2-‑7 complexes with mutations eliminating DDK phosphorylation sites on the N-terminal tails of either Mcm4 or Mcm6. We have tested these mutant proteins in the single-molecule assay for Cdc45 association and CMG formation (modified Figure 6). Importantly, for both mutants we observe a corresponding reduction in both CMG formation and the multiplicity of Cdc45 association.

Although the Mcm4 and Mcm6 phosphorylation mutations we tested are known to impact cell viability from previous studies (Randell et al., 2010), they had not been tested for their impact on replication origin function. To address this possibility, we measured initiation efficiency in vivo from early initiating origins of replication using strains containing these Mcm2-7 variants (new Figure 7). Importantly, we find that these mutations reduce origin replication efficiency in a manner that reflects their impact on CMG formation.

3) Figure 1B. Can the authors explain why the number of molecules observed in DDK/Sld3/lambda PP is >13 fold lower than n in DDK/Sld3?

Two aspects of the experiments contribute to this difference. First, the experiments that did not involve lambda PP report the results of several experiments that were combined, increasing n values. The experiments with DDK are the results of a single experimental repetition (we now indicate the number of contributing experiments in the legend for Figure 1B). Second, the manipulations required to dephosphorylate and phosphorylate the Mcm2-7 complexes led to less efficient Mcm2-7 loading in the lambda phosphatase experiments. Although we initially thought that dephosphorylation of Mcm2-7 might reduce its ability to be loaded onto origin DNA, subsequent experiments using ensemble assays showed no difference in loading of dephosphorylated Mcm2-7 (see Author response image 1). Importantly, although the n values are lower in these experiments, the differences that we emphasize are clearly statistically significant.

**Author response image 1. respfig1:** Lambda phosphatase (λPP) does not affect Mcm2-7 loading. Ensemble Mcm2-7 loading assays were conducted in the presence and absence of λPP. A loading control in the absence of Cdc6 is included as well as inputs for Mcm2-7. Note that the Mcm2 protein migrates more slowly after dephosphorylation.

4) Figure 2. In panel A the intensities fluctuation is high, suggesting that up to four Cdc45 molecules bind at a time. Conversely, when SDPGC is present variation is greatly reduced and only up to 1 or 2 molecules of Cdc45 bind. Do SDPGC prevent non-specific interactions? This could be discussed.

Figure 2C shows that when the complete sets of experimental records are analyzed, essentially all Mcm2-7-bound DNAs have either zero, one, or two dye-labeled Cdc45 molecules bound at any given point in time. This is the case whether or not the SDPGC proteins are present; including those proteins caused only a minor increase in the fraction of ones and two and a corresponding minor decrease in the faction of zeros. The two Cdc45 fluorescence records chosen as examples (in Figure 2A and B) are both good illustrations of these features of the complete set of DNA records.

The reviewer correctly perceives that the Cdc45 intensity in Figure 2A reaches higher values than those in 2B, but the difference in peak intensity is not caused by a difference in the *number* of dye-labeled Cdc45 molecules bound -- that number is zero, one, or two at nearly all time points in both records. Instead, inspection of the records reveals that the difference is in the *sizes* of the stepwise increases/decreases in fluorescence intensity corresponding to binding/dissociation of a single dye molecule (examples of these steps are marked by arrows in the figures.) This unitary step size is ~20,000 in Figure 2A but only ~10,000 in Figure 2B. In these experiments, such random variation in step size from one DNA to the next is often observed, even between different DNA molecules in the same experimental sample (for example, compare the first and third plots in Figure 2—figure supplement 1), and on average the step size is not appreciably larger in the –SDPGC sample than in the + SDPGC sample (~13,000 vs. ~11,000; see the *s* parameter in rows 1 and 2 of Supplementary file 1A). Thus, the difference noted by the reviewer is a property of the two example records chosen but not of the dataset as a whole, the properties of which are displayed in Figure 2C.

5) Do the authors only analyse loaded MCM molecules that are double-hexameric? See the representative photobleaching curve in Figure 2A. Can the authors explain whether or not only particles that contain 2 fluorescently labelled MCM complexes are included in the analysis?

All of the experiments include a high-salt wash after the DDK phosphorylation step to ensure that only loaded Mcm2-7 molecules are involved in the helicase activation reaction (the presence of CDK during the activation step also ensures that no new loading is occurring at this stage). This results in the majority of the Mcm2-7 complexes that are present during the helicase activation portion of the reaction being double hexamers (Ticau et al., 2015). We note that the traces in 2A and 2B (for Mcm2-7) show evidence for two fluorescent molecules (one that photobleaches during the observations and the second that does not) and these records were chosen because this pattern of photobleaching of Mcm2-7 was the most common. Because of incomplete labeling of Mcm2-7 (we measured labeling of Mcm2-7 to be 74 ± 5% for Mcm2-7^4SNAP549^), we expect to see a subset of loaded Mcm2-7 complexes to have only one of the two Mcm2-7 double hexamers to be labeled. For that reason, we chose to analyze all the molecules that contain a detectable Mcm2-7. In rare cases neither Mcm2-7 will be labeled and this situation likely explains GINS associations that occur in the absence of a labeled Mcm2‑7 (e.g. Figure 5—figure supplement 1).

Because it was possible that loaded single Mcm2-7 molecules contribute to our findings, we have analyzed a set of molecules from the high DDK Cdc45 loading that we can definitively say are double-hexamers based on fluorescent intensity records and compared them to the analysis shown in the paper (Author response image 2). This represents 51% of the total DNA molecules with associated Mcm2-7. We find that we observe the same distribution of Cdc45 binding under these conditions consistent with the inclusion of all the Mcm2-7 associated molecules in our analysis throughout the paper.

**Author response image 2. respfig2:** Cdc45 binding distribution is unchanged when using an Mcm2-7 double-hexamer (DH) selection. (**A**) Cdc45^SORT649^ fluorescence**-**intensity histograms for all detectable Mcm2-7^4SNAP549^. (**B**) Cdc45^SORT649^ fluorescence**-**intensity histograms for only Mcm2-7^4SNAP549^ DHs.

6) Have the authors attempted to omit Mcm10 from the CMG formation assay? And could they observe any change in dye emission?

The only experiments in the paper that include Mcm10 are the helicase activity experiments shown in Figure 1C and D. As indicated by the illustrations of the order of protein addition provided with each new reaction we present, all of the CMG formation assays were performed in the absence of Mcm10 and RPA to focus on the CMG formation rather than the actively unwinding CMG. Although it will be interesting to develop assays to monitor the changes to Mcm2-7, GINS, or Cdc45 in the future, such experiments are beyond the scope of this study.

7) In the subsection “Only a subset of the multiple Mcm2-7-bound Cdc45 proteins form CMGs”, the authors conclude that the two CMGs can assemble independently at each MCM DH. The statistical data for this conclusion are not convincing. Can the authors exclude that the 41% observed vs. 57% expected MCM DHs with two salt-resistant labeled Cdc45 molecules is not due to an error in the estimation of the Cdc45 labeling efficiency? Also, in the case of 1.3 nM DDK only 7 molecules were analyzed, which seems too low a number to support any statistically significant conclusion. As the question of coordination between CMG assembly at the MCM DH is mechanistically very important, the authors may want to consider to tone down the conclusion that CMGs can assembly independently or provide additional data in support of their conclusion.

We agree with the reviewers that this is an important mechanistic point. The Cdc45 fraction labeled is measured in a single-molecule experiment (see Materials and methods) in which we believe the dominant source of uncertainty to be the counting statistics, rather than any systematic error in the measurement. The statistical error in the measurement of the labeling efficiency is included in the standard errors reported for the fraction of labeled complexes (0.57 ± 0.09) predicted if all complexes have two salt-stable Cdc45 molecules. This differs significantly from the observed fraction of labeled complexes (0.41 ± 0.05) in the 6.5 nM DDK experiment. In the 1.3 nM DDK experiment the uncertainty due to the small number of observations is quantified by the large relative standard error in the measurement (0.14 ± 0.13). Despite this, it is unambiguous that this number is significantly different from the 0.57 ± 0.09 predicted for the coordinated mechanism.

While we agree that these data are not definitive evidence for an uncoordinated model, we do think they are strong enough to merit description and discussion. We believe our current wording reflects the lack of definitive evidence: “These findings *suggest* that CMG formation on the two Mcm2-7 complexes in each double hexamer *can* occur independently.” If the reviewers feel strongly, we could consider an alternate wording such as: “These data are consistent with a model in which CMG formation on the two Mcm2-7 complexes in each double hexamer can occur independently.”

8) The authors demonstrate that GINS binding to MCMs is dependent on Sld3/7 and Cdc45. Is GINS binding dependent on Sld2, Dpb11, Pol-epsilon, and CDK as well? Does elimination of any one of these factors differentially affect GINS recruitment and CMG formation? This could point to the mechanism of CtG to CMG transformation. The authors are well-positioned to address this question and may have done so already.

We have added an additional experiment along these lines. We have assessed the impact of eliminating Dpb11 (Figure 5A and Figure 5—figure supplement 2) on GINS binding events. Consistent with a role of Dpb11 in GINS initial Mcm2-7 recruitment, we observe nearly a complete loss of GINS binding in the absence of this protein. Interestingly, we observe a much stronger effect of Dpb11 elimination on GINS binding events than we observe for elimination of Cdc45 or Sld3/7. We have added a brief discussion as to why this might be the case (Discussion).

9) Figure 4—figure supplement 2B: How was this experiment performed. The text states that protein was stained with Krypton protein stain. Where are the other MCM subunits then? Or was this experiment performed with isolated Mcm4 tail? Please clarify.

We apologize for the lack of clarity regarding this experiment. Although the experiment looking at Mcm6 was done with the complete Mcm2-7 complex, the experiment with Mcm4 was performed with the tail of Mcm4 expressed in the absence of the rest of the Mcm2-7 complex. We have clarified this distinction in the text, in the legend of Figure 4—figure supplement 2, and in the Materials and methods.